# Global organic and inorganic aerosol hygroscopicity and its effect on radiative forcing

Mira L. Pöhlker [1,2,3] ✉, Christopher Pöhlker [1], Johannes Quaas [2], Johannes Mülmenstädt[2,17], Andrea Pozzer [4,5], Meinrat O. Andreae [6,7], Paulo Artaxo [8], Karoline Block[2], Hugh Coe [9], Barbara Ervens[10], Peter Gallimore[9], Cassandra J. Gaston[11], Sachin S. Gunthe[12,13], Silvia Henning [3], Hartmut Herrmann [14], Ovid O. Krüger[1], Gordon McFiggans [9], Laurent Poulain[14], Subha S. Raj[1,12], Ernesto Reyes-Villegas [9,18], Haley M. Royer[11], David Walter [1,15], Yuan Wang[3,16] & Ulrich Pöschl[1]

The climate effects of atmospheric aerosol particles serving as cloud condensation nuclei (CCN) depend on chemical composition and hygroscopicity, which are highly variable on spatial and temporal scales. Here we present global CCN measurements, covering diverse environments from pristine to highly polluted conditions. We show that the effective aerosol hygroscopicity, $\kappa$, can be derived accurately from the fine aerosol mass fractions of organic particulate matter ($\epsilon_{org}$) and inorganic ions ($\epsilon_{inorg}$) through a linear combination, $\kappa = \epsilon_{org} \cdot \kappa_{org} + \epsilon_{inorg} \cdot \kappa_{inorg}$. In spite of the chemical complexity of organic matter, its hygroscopicity is well captured and represented by a global average value of $\kappa_{org} = 0.12 \pm 0.02$ with $\kappa_{inorg} = 0.63 \pm 0.01$ as the corresponding value for inorganic ions. By showing that the sensitivity of global climate forcing to changes in $\kappa_{org}$ and $\kappa_{inorg}$ is small, we constrain a critically important aspect of global climate modelling.

The radiative energy budget of the Earth is strongly influenced by atmospheric aerosol particles, which scatter and absorb solar radiation, act as nuclei in the formation of cloud droplets and ice crystals, and cause a variety of rapid adjustments in environmental variables[1,2].

In our understanding of the climate system and our ability to model its human-driven change, the roles of aerosols, such as aerosol–radiation interactions (ari) and particularly the highly dynamic aerosol–cloud interactions (aci), have remained strikingly uncertain[3,4]. The aerosol

[1]Multiphase Chemistry Department, Max Planck Institute for Chemistry, 55128 Mainz, Germany. [2]Faculty of Physics and Earth Sciences, Leipzig Institute for Meteorology, Leipzig University, 04103 Leipzig, Germany. [3]Atmospheric Microphysics Department, Leibniz Institute for Tropospheric Research, 04318 Leipzig, Germany. [4]Atmospheric Chemistry Department, Max Planck Institute for Chemistry, 55128 Mainz, Germany. [5]Climate and Atmosphere Research Center, The Cyprus Institute, 2121 Nicosia, Cyprus. [6]Biogeochemistry Department, Max Planck Institute for Chemistry, 55128 Mainz, Germany. [7]Scripps Institution of Oceanography, University of California San Diego, La Jolla, CA 92037, USA. [8]Instituto de Física, Universidade de São Paulo, São Paulo, Brazil. [9]Department of Earth and Environmental Sciences, School of Natural Sciences, University of Manchester, Manchester, UK. [10]Université Clermont Auvergne, CNRS, Institut de Chimie de Clermont-Ferrand, 63000 Clermont-Ferrand, France. [11]Department of Atmospheric Sciences, Rosenstiel School of Marine and Atmospheric Science, University of Miami, Miami, FL 33149-1031, USA. [12]Environmental Engineering Division, Department of Civil Engineering, Indian Institute of Technology Madras, Chennai, India. [13]Center for Atmospheric and Climate Sciences, Indian Institute of Technology Madras, Chennai, India. [14]Atmospheric Chemistry Department, Leibniz-Institute for Tropospheric Research, 04318 Leipzig, Germany. [15]Climate Geochemistry Department, Max Planck Institute for Chemistry, 55128 Mainz, Germany. [16]Collaborative Innovation Center for Western Ecological Safety, Lanzhou University, 730000 Lanzhou, China. [17]Present address: Pacific Northwest National Laboratory, Richland, WA 99354, USA. [18]Present address: School of Engineering and Sciences, Tecnologico de Monterrey, Guadalajara 45201, Mexico. ✉e-mail: poehlker@tropos.de

hygroscopicity and, thus, the water content of the particles in response to the atmospheric conditions, influence the aerosol-related effective radiative forcing (ERFaer) in the climate system via two mechanisms: First, more hygroscopic aerosol particles grow to larger diameters at relative humidity < 100%, leading to a greater scattering cross section and thus stronger radiative forcing by aerosol–radiation interactions (RFari). Second, particles of the same size but higher hygroscopicity act more readily as cloud condensation nuclei (CCN), which activate into cloud droplets, leading to a larger cloud droplet number, longer-lived clouds and stronger forcing by aerosol–cloud interactions (RFaci)[5], and, together with the cloud adjustments, this entails a stronger ERFaer.

The Köhler equation

$$s = a_w \cdot \text{Ke} \qquad (1)$$

represents the dependence of $s$—which is the ratio of equilibrium water vapour pressure over a curved solution to that above a flat, pure-water surface—on the Raoult and Kelvin effects[6]. The equation combines the solute (Raoult) term, which quantifies the decreased water activity, $a_w$, leading to a decreased water vapor pressure above the water surface as a function of particle size and chemical composition, and the curvature (Kelvin) term, Ke, which quantifies the water vapor pressure enhancement above a curved surface as a function of surface tension. Equation (1) is of fundamental importance in cloud microphysics, as it thermodynamically describes the activation of CCN into cloud droplets at a given water vapor supersaturation, $S = s - 1$. Petters and Kreidenweis[7] introduced the hygroscopicity parameter, $\kappa$, to parameterize $a_w$ of a solution droplet according to

$$a_w = \left(1 + \kappa \frac{V_s}{V_w}\right)^{-1} \qquad (2)$$

where $V_s$ and $V_w$ are the volumes of the dry solute and pure water, respectively. In combination with particle size and number concentration, $\kappa$ has been very useful to predict CCN number concentrations in aerosol populations[8–10]. Experimentally, $\kappa$ has been retrieved via two different strategies: (i) from ambient measurements of entire aerosol populations, resulting in overall effective $\kappa$ values, typically

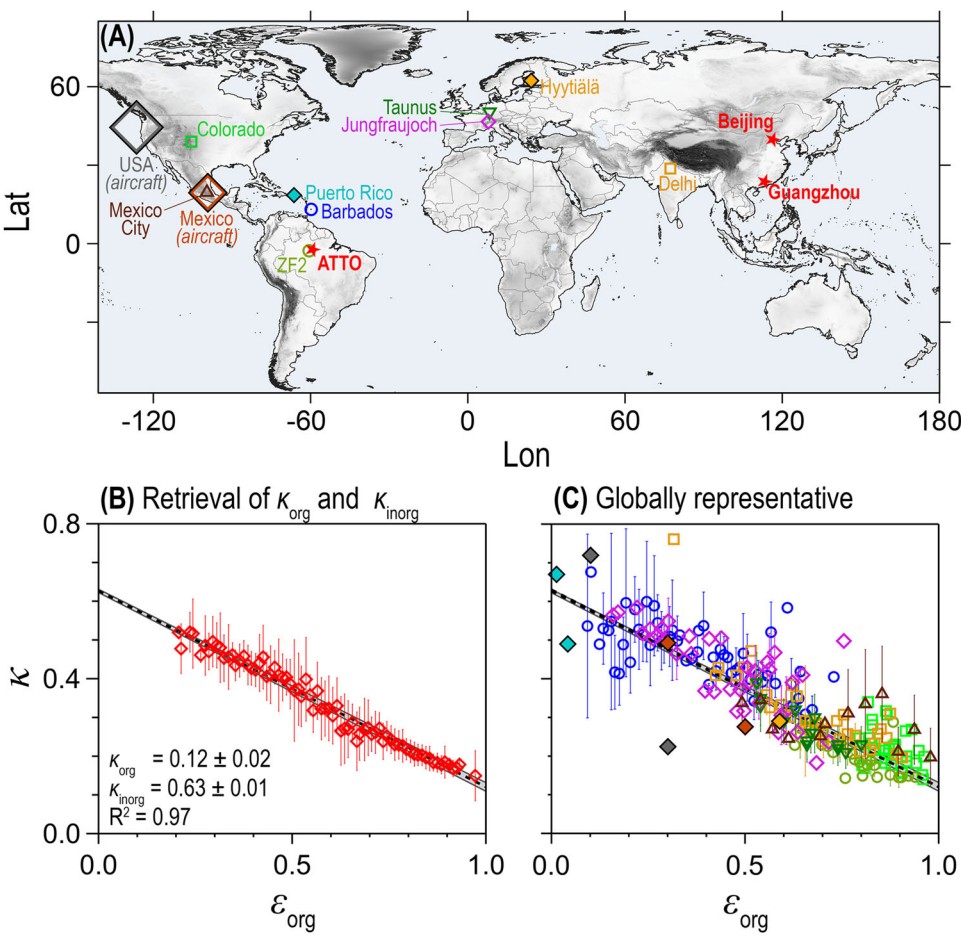

**Fig. 1 | Retrieval of globally representative aerosol hygroscopicity parameters for organic particulate matter, $\kappa_{org}$, and inorganic ions, $\kappa_{inorg}$, based on cloud condensation nuclei measurements worldwide.** Retrieval of $\kappa_{org} = 0.12 \pm 0.02$ and $\kappa_{inorg} = 0.63 \pm 0.01$ from linear bivariate regression fits of experimentally derived organic mass fraction, $\epsilon_{org}$, and hygroscopicity parameter, $\kappa$, from the Amazon, Beijing, and Guangzhou in (**B**), along with further campaign data sets in (**C**), showing that $\kappa_{org}$ and $\kappa_{inorg}$ are representative for organic and inorganic aerosols under continental and marine conditions worldwide. The global map in (**A**) with topography in gray shows all measurement locations relevant for this study. The retrieval in (**B**) is based on the three extended campaign datasets: Program of

Regional Integrated Experiments of Air Quality over the Pearl River Delta (PRIDE-PRD2006) in Guangzhou[59], Campaign of Air Quality Research in Beijing (CAR-EBeijing-2006)[61], and at the Amazon Tall Tower Observatory (ATTO)[19] (Table 2). The regression fit is shown as black dashed lines with grey shading as uncertainty of the fit. The regression line from (**B**) is repeated in (**C**) with previously published[10,16,20–25,70,73] as well as new datasets (Table 2). Markers represent geometric mean of $\epsilon_{org}$ bins and error bars standard deviation. The colors and shapes of the markers in (**C**) as well as in Fig. S3 are identical to relate the data sets to the corresponding sites in (**A**).

ranging from -0.1 to -0.9, and (ii) from laboratory experiments yielding compound-specific $\kappa$ values ranging from -0.02 to -0.4 for organic and from -0.3 to -0.9 for inorganic reference substances[7,11,12]. Typical inorganics are salts comprising sulfate ($SO_4^{2-}$), nitrate ($NO_3^-$), ammonium ($NH_4^+$), or chloride ($Cl^-$). For the calculation of $\kappa$ as an effective parameter, often the surface tension of pure water has been assumed, as is also done here.

The $\kappa$ of a mixture of chemical components, $i$, in an aerosol population is given by the additive mixing rule

$$\kappa = \sum_i \epsilon_i \kappa_i \tag{3}$$

which includes the dry component volume fraction $\epsilon_i = V_{si}/V_s$ with $V_{si}$ as the volume of the individual components and $V_s$ as the total particle volume[7,13,14]. This formula is based on the Zdanovskii-Stokes-Robinson (ZSR) approach assuming additive water uptake of individual components in mixtures[15]. Aerosol chemical composition is measured routinely by aerosol mass spectrometers (AMS), providing time series of the quantitative mass concentrations, $m$, of the major aerosol constituents organics (Org), $SO_4^{2-}$, $NO_3^-$, $NH_4^+$, and $Cl^-$. While the mass fraction of inorganic ions is usually limited to a few well characterized compounds, the organic mass fraction might comprise thousands of different molecules, of which most are not fully identified. Standardized AMS observations worldwide have improved the understanding of global aerosol composition profoundly[11]. This has particularly helped to constrain the ambient aerosol hygroscopicity, since the AMS-derived data on aerosol composition can be directly linked to an ambient $\kappa$ using the simplified version of Eq. (3)

$$\kappa = \epsilon_{org}\kappa_{org} + \epsilon_{inorg}\kappa_{inorg} \tag{4}$$

with the organic mass fraction $\epsilon_{org} = m_{org}/m_{total}$, the inorganic mass fraction $\epsilon_{inorg} = (m_{total} - m_{org})/m_{total}$, and the total aerosol mass concentration $m_{total} = m_{org} + m_{SO_4^{2-}} + m_{NO_3^-} + m_{NH_4^+} + m_{Cl^-}$[16]. Note that Eq. (4) is an approximation and assumes that the inorganics are composed solely of hygroscopic salts. As mineral components and elemental carbon are not detected by the AMS, the presence of non- or weakly-hygroscopic compounds as minor constituents is subsumed into the deduced value of $\kappa_{org}$ and/or $\kappa_{inorg}$. The upper size limit of AMS measurements (i.e., 50% transmission) and therefore the data used in Eq. (4) is typically around 600 nm[17]. Further, whereas Eq. (3) is based on volume fractions, Eq. (4) uses mass fractions, which is an approximation. This is justified here as the densities, $\rho$, are sufficiently similar (i.e., $\rho_{(NH_4)_2SO_4^{2-}} = 1.77\,g\,cm^{-3}$, $\rho_{NH_4NO_3} = 1.72\,g\,cm^{-3}$, $\rho_{org,average} = 1.4\,g\,cm^{-3}$[18]), especially for high $\epsilon_{org}$ (i.e., $\gtrsim 0.75$). For lower $\epsilon_{org}$ (i.e., $\approx 0.25$), however, the difference between volume and mass fractions can be up to 15% (with volume fractions being larger), which has to be considered in the interpretation of the results.

Equation (4) has been applied in both directions: either as a retrieval of $\kappa_{org}$ and $\kappa_{inorg}$ based on ambient measurements of an overall $\kappa$ and aerosol chemical composition or as an application of previously derived $\kappa_{org}$ and $\kappa_{inorg}$ values to parameterize the effective aerosol hygroscopicity, e.g., for the purpose of radiative transfer calculations. In fact, aerosol–climate models use Eq. (3) to parameterize the aerosol hygroscopicity through species-specific $\kappa$ values (see Methods). Several retrievals of $\kappa_{org}$ and $\kappa_{inorg}$ have been reported, from remote tropical forests to moderately or even heavily polluted urban areas (Table S1). These campaign-specific results show large ranges in $\kappa_{org}$ from 0.04 to 0.25 and for $\kappa_{inorg}$, from 0.46 to 0.71, and further emphasize the need of a robust and accurate representation of the aerosol hygroscopicity in aerosol–climate models. This raises two fundamental questions:

1. Is there a pair of robust $\kappa_{org}$ and $\kappa_{inorg}$ values that is globally representative of aerosol water contents above and below water

saturation and, thus, a suitable choice for general application in aerosol–climate models?
2. How sensitive are the model predictions in terms of CCN concentrations as well as direct and indirect radiative forcing to the choice of $\kappa_{org}$ and $\kappa_{inorg}$?

This study addresses both questions. First, we conducted a systematic retrieval of $\kappa$ by merging extensive data sets from AMS and size-resolved CCN measurements obtained in contrasting environments (Fig. 1A) to calculate a pair of broadly representative $\kappa_{org}$ and $\kappa_{inorg}$ values. Secondly, we applied this optimized retrieval along with other pairs of $\kappa_{org}$ and $\kappa_{inorg}$ in the ECHAM–HAM aerosol–climate model to test the corresponding sensitivity of CCN concentrations and radiative forcing.

## Results and discussion
### Retrieval of globally representative $\kappa_{org}$ and $\kappa_{inorg}$ values

Figure 1B shows how $\kappa_{org}$ and $\kappa_{inorg}$ were retrieved through a linear bivariate regression fit of measured $\epsilon_{org}$ and $\kappa$ data, combining contrasting data from remote rain forest (Amazon) to urban-polluted air (Chinese megacities). The extrapolation of the fit to $\epsilon_{org} = 0$ yields $\kappa_{inorg}$ and the extrapolation to $\epsilon_{org} = 1$ yields $\kappa_{org}$[16,19]. The optimized retrieval shows a clearly linear relationship, consistent with Eq. (4), as well as a remarkably tight scatter plot and corresponding low uncertainty. It yields $\kappa_{org} = 0.12 \pm 0.02$ and $\kappa_{inorg} = 0.63 \pm 0.01$ as robust values, constrained by a wide distribution of $\epsilon_{org}$ data points from -0.2 to -1.0.

Figure 1C underlines how robustly and broadly our optimized retrieval represents the average aerosol hygroscopicity worldwide by means of a good agreement between the regression line from Fig. 1B and measurement data from 11 additional field campaigns, including remote background locations (i.e., boreal forest[10,20], semi-arid Northern American forest[21], marine[22], alpine[23]), rural locations (in Europe[24] and North America[25]) as well as strongly polluted regions (i.e. Indo-Gangetic Plain and central Mexico[25]). The individual data sets are shown in Fig. S3. The data from the marine background site on Barbados as well as a polluted site in the Indo-Gangetic Plain (Delhi) are of particular significance as only few aerosol and CCN data exist from these locations. Note that the aerosols with relatively high fractions of refractory particles (i.e., sea salt from marine sites and mineral dust at an alpine location) follow the regression line fairly well, even though a certain fraction is not quantitatively evaporated in the AMS and, thus not included in $\epsilon_{inorg}$. The good agreement can be explained as it is primarily the soluble matter in the dust aerosol that affects the CCN population, whereas large dust particles, which could nucleate droplets, are not abundant enough to affect $\kappa$ significantly[26]. For marine aerosols, it has been well documented by now that in most cases a mixture of secondary sulfate and organic aerosols—both quantified by the AMS—accounts for most of the CCN[27,28]. Note here that sea spray and mineral dust aerosol populations typically reach far into the supermicron size range, which implies that the AMS measurement with its upper cut-off size at about 600 nm—defined here as 50% transmission[17]—excludes significant parts of both aerosol components. Since primary and secondary organic aerosols (POA and SOA) cannot be distinguished by the AMS and thus are combined in the measured organic mass concentration, we recommend to use $\kappa_{org} = 0.12$ for both SOA and POA, e.g., in model parametrizations. Following the approach in Gunthe et al.[16] and interpreting $\kappa_{org} = 0.12$ as an "effective Raoult parameter", we calculated an effective average molecular mass of the organic compounds of $M_{org} = 212 \pm 42\,g\,mol^{-1}$, assuming a density of $1.4\,g\,cm^{-3}$[18], in good agreement with previous studies[16,29,30].

Overall, the good agreement among all data sets in Fig. 1C provides strong evidence that our optimized retrieval of $\kappa_{org} = 0.12 \pm 0.02$ and $\kappa_{inorg} = 0.63 \pm 0.01$ is a widely applicable representation of the average effective hygroscopicity of aerosols consisting of mixtures of organic and inorganic components over oceans and continents

worldwide. This mixing state of the aerosol components affects ari and aci and its representation in models is associated with large uncertainties[31–33]. Generally, an aerosol populations tends to become more and more internally mixed with atmospheric residence time, even within tens of kilometers or after few hours of transport[34,35]. At the same time, studies suggest that the mixing state of atmospheric aerosols is not fully represented by either the assumption or parameterization of a completely external mixture or internal mixture[36,37]. Our parameterization of $\kappa_{org}$ and $\kappa_{inorg}$ is based on a wide variety of atmospheric conditions at different locations and, therefore, represents an atmospherically relevant distribution of prevalent aerosol mixing states. At the same time, special aerosol populations, such as freshly emitted and externally mixed particles, might deviate from the average retrieval. Further note that AMS measurements barely account for the chemical complexity of the organic fraction as they, for instance, do not differentiate between inorganic sulfate or nitrate vs

sulfate or nitrate fragments emerging from organosulfates or organonitrates. Vogel et al.[38] argue that this effect causes a systematic underestimation of $\epsilon_{org}$ for significant levels of organosulfates or organonitrate (Fig. S4), which after correction is still consistent with the linear relationship reported here. Nevertheless, this aspect is worth addressing in follow-up studies in (nitrooxy)-organosulfate-rich environments with targeted instrumentation.

## Climate model sensitivity to changes in $\kappa_{org}$ and $\kappa_{inorg}$

Changes in the aerosol hygroscopicity affect the growth of the particles in the subsaturated regime and therefore RFari as well as their activation to cloud droplets under supersaturated conditions and therefore RFaci. Accordingly, an improved determination of $\kappa$ better constrains the aerosol effects on the radiative forcing and therefore the climate. The aerosol–climate model ECHAM–HAM, which is the ECHAM atmospheric general-circulation model[39] coupled to the HAM aerosol module[40], was used to estimate the sensitivity of the radiative forcing in climate models to changes in aerosol hygroscopicity. The model parameterizes the overall $\kappa$ through Eq. (3) based on compound-specific $\kappa$ values for sea salt, black carbon, mineral dust, organic carbon, and sulfate aerosols (details in experimental section). Replacing the model's standard values, which are 0.06 for organics and 0.6 for inorganics, by our optimized values $\kappa_{org} = 0.12$ and $\kappa_{inorg} = 0.63$ yields absolute changes in the global mean radiative forcing at the top of the atmosphere of $\Delta$RFari $= -0.011\,W\,m^{-2}$ and $\Delta$ERFaer $= -0.026\,W\,m^{-2}$ (Table 1). This corresponds to relative changes in RFari of 8% for a replacement of $\kappa_{inorg}$ and 1% for a replacement of $\kappa_{org}$ as well as a relative change in ERFaer of 1% for a replacement of both, $\kappa_{inorg}$ and $\kappa_{org}$. The absolute changes are close to the range of the model variability for the 30-year model runs. Further note the $\Delta$RFari and $\Delta$ERFaer from ECHAM–HAM can be rather considered as lower limit values and corresponding results from other climate models might be higher. Myhre et al.[41] showed in a model comparison study that ECHAM-HAM yields a global mean anthropogenic RF (all-sky) of

**Table 1 | Global mean values for radiative forcing from aerosol-radiation interactions (RFari) and effective radiative forcing from aerosol-radiation and aerosol-cloud interactions with rapid adjustments (ERFaer) at the top of the atmosphere, as net flux perturbations in the solar and terrestrial spectra, calculated with the Ghan[96] method for the various experiments with different prescribed $\kappa$ parameterizations to assess the corresponding sensitivity of ECHAM–HAM**

| Model experiment | $\kappa_{inorg}$ | $\kappa_{org}$ | RFari [W m$^{-2}$] | ERFaer [W m$^{-2}$] |
|---|---|---|---|---|
| Reference case | 0.60 | 0.06 | − 0.109 ± 0.004 | − 1.757 ± 0.018 |
| $\kappa_{OC} = \kappa_{org}$ | 0.60 | 0.12 | − 0.110 ± 0.004 | − 1.748 ± 0.019 |
| $\kappa_{SU} = \kappa_{inorg}$ | 0.63 | 0.06 | − 0.118 ± 0.004 | − 1.771 ± 0.020 |
| $\kappa_{OC} = \kappa_{org}$ & $\kappa_{SU} = \kappa_{inorg}$ | 0.63 | 0.12 | − 0.120 ± 0.004 | − 1.773 ± 0.019 |

Uncertainty in RFari and ERFaer represents the standard deviation of the multi-year simulations.

**Table 2 | Overview of data sets from different field experiments used in Fig. 1**

| # | campaign / site name & atm. conditions | lat, long | time | Reference |
|---|---|---|---|---|
| 1 | Amazon Tall Tower Observatory (ATTO), BRA, remote tropical rain forest | 2.143° S, 59.000° W | Mar 2014–Feb 2015 | 9,19,52 |
| 2 | PRIDE-PRD2006, Guangzhou, CHN, rural site near megacity | 23.548° N, 113.066° E | Jul 2006 | 58,59 |
| 3 | CAREBeijing-2006, Beijing, CHN, suburban site near megacity | 39.515° N, 116.305° E | Aug & Sep 2006 | 60,61 |
| 4 | Delhi-2018 campaign, IND, strongly polluted Indo-Gangetic plain | 28.588° N, 77.221° E | Feb & Mar 2018 | 62–64 |
| 5 | Barbados Atmospheric Chemistry Observatory (BACO), BRB, marine background with episodes of African long-range transport | 13.165° N, 59.432° W | Jan & Feb 2020 | 66 |
| 6 | Puerto Rico Aerosol Cloud Interaction Study (PRACS), PRI, clean marine air with some pollution | 18.390° N, 65.619° W | Dec 2004 | 22 |
| 7 | Feldberg Aerosol Characterization Experiment (FACE-2005), Taunus Observatory, DEU, rural background | 50.223° N, 8.447° E | Jun–Jul 2005 | 24 |
| 8 | Cloud and Aerosol Characterization Experiment (CLACE-06), Jungfraujoch, CHE, high alpine site probing free troposphere background | 46.550° N, 7.983° E | Mar 2007 | 23 |
| 9 | Amazonian Aerosol Characterization Experiment (AMAZE-08), ZF2 site, BRA, tropical rain forest | 2.594° S, 60.209° W | Mar 2008 | 16,67 |
| 10 | European Integrated project on Aerosol Cloud Climate and Air Quality Interactions (EUCAARI) campaign, SMEAR-II site, Hyytiälä, FIN, boreal forest | 61.850° N, 24.283° E | Feb 2009–Apr 2012 | 10,68 |
| 11 | Megacities Impact on Regional and Global Environment (MIRAGE) Experiment, aircraft, MEX | airborne | Mar 2006 | 25,69 |
| 12 | MILAGRO and MIRAGE, Mexico City, MEX, megacity pollution | several sites | Mar 2006 | 69,70 |
| 13 | INTEX-B, aircraft, USA | airborne | Apr–May 2006 | 25,71 |
| 14 | Taunus Observatory, DEU, rural background | 50.223° N, 8.447° E | Aug 2012 | 38 |
| 15 | European Integrated project on Aerosol Cloud Climate and Air Quality Interactions (EUCAARI) 2007 campaign, SMEAR-II Hyytiälä, FIN, boreal forest | 61.850° N, 24.283° E | Mar–May 2007 | 72 |
| 16 | BEACHON, Colorado, USA, Rocky Mountains semi-arid forest | 38.64° N, 105.11° W | Mar 2010–Feb 2011 | 21,73 |

Information has been obtained from the original studies, where further details can be found.

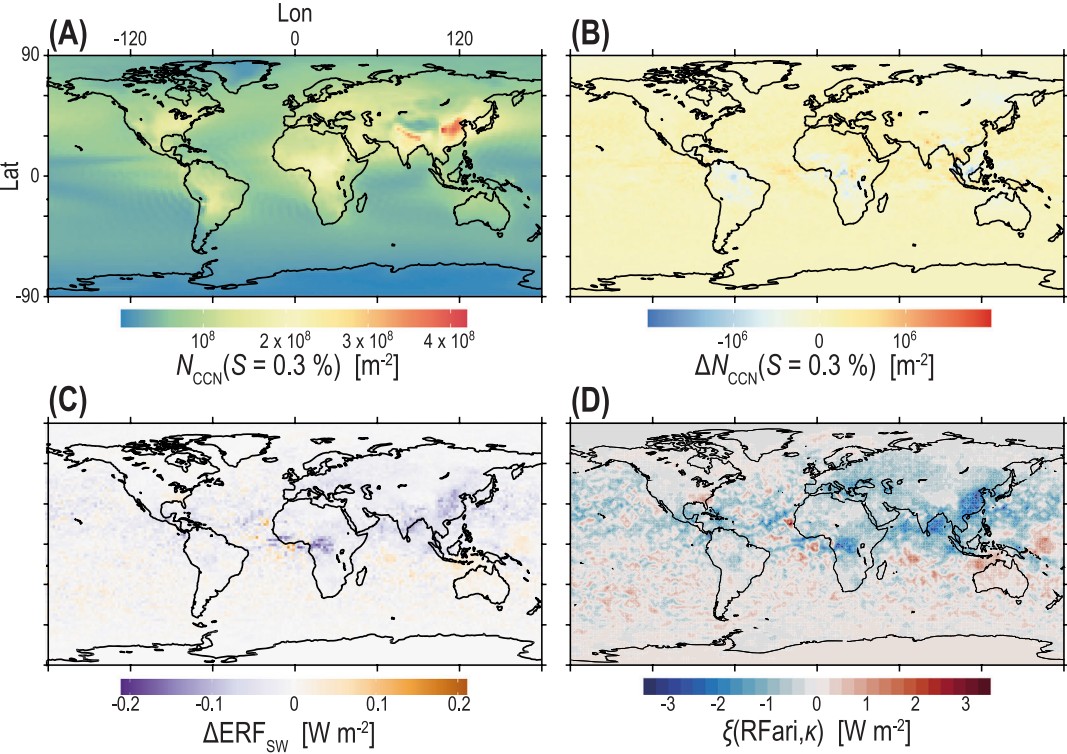

**Fig. 2 | Sensitivity of modelled global cloud condensation nuclei concentration and climate forcing to changes in $\kappa_{\text{org}}$ and $\kappa_{\text{inorg}}$.** Predicted changes in the global distribution of the cloud condensation nuclei (CCN) number concentration and associated effective radiative forcing in the short-wave, SW (solar) range, when the default $\kappa$ values of a global climate in the model ($\kappa_{\text{OC}} = 0.06$ and $\kappa_{\text{SU}} = 0.60$), are replaced by our newly derived values $\kappa_{\text{org}} = 0.12$, $\kappa_{\text{inorg}} = 0.63$. **A** Annual-average column CCN number concentration at $S = 0.3\%$, $N_{\text{CCN}}(S = 0.3\%)$ using the default $\kappa$ values. **B** Predicted difference in $N_{\text{CCN}}(S = 0.3\%)$ when the default $\kappa$ values are replaced by the newly derived ones. **C** Change in the effective radiative forcing in the SW range ($\Delta\text{ERF}_{SW}$) at top-of-atmosphere (TOA) as the difference in ERF between pairs of simulations with the two different $\kappa$ parameter choices. $\Delta\text{ERF}_{SW}$ has been estimated from the difference in TOA SW radiative flux between simulations with present-day (PD) and pre-industrial (PI) aerosol emissions and primarily reflects here the change in the radiative forcing from the aerosol-radiation interaction (RFari). **D** Sensitivity of the RFari to the combined change in $\kappa_{\text{org}}$ and $\kappa_{\text{inorg}}$.

−0.15 W m$^{-2}$, in comparison to the 16-model mean of − 0.27 ± 0.15 W m$^{-2}$ (±one standard deviation). Along these lines, the RFari of ≃−0.11 W m$^{-2}$ derived here from ECHAM-HAM (Table 1) is much less than the value of −0.3 W m$^{-2}$ given in the recent report of the Intergovernmental Panel on Climate Change (IPCC).

In principle, the aerosol effects through ari and aci could be treated approximately additively, which allows to retrieve RFaci as ERFaer − RFari from Table 1. We refrain from reporting these values here, however, as this approach does not account for the various adjustments in the system upon ari or aci changes and also cannot be obtained directly from the model. In terms of indirect effects, the replacement of $\kappa$ yields the geographic distributions of the changes in CCN concentrations ($\Delta N_{\text{CCN}}$) and in the effective radiative forcing ($\Delta\text{ERF}_{SW}$) shown in Fig. 2B, C. These changes in $\Delta N_{\text{CCN}}$ and $\Delta\text{ERF}_{SW}$ are comparatively small, which is not surprising since out of the aerosol-related factors that control the CCN spectrum—which are total number concentration, particle size, and particle hygroscopicity—the sensitivity to number is the largest. Moreover, the simplified Köhler equation (Eq. (9)) underlines that the exact value of $\kappa$ is generally the least important determinant of CCN activity as $S$ varies with $D$ to the power of three and just linearly with $\kappa$[42]. The largest $\Delta N_{\text{CCN}}$ and $\Delta\text{ERF}_{SW}$ values observed in Fig. 2B,C were found in regions with a high organic aerosol burden and the corresponding values (filtered for high OC) of $\Delta\text{RFari} = -0.049$ W m$^{-2}$ and $\Delta\text{ERFaer} = -0.149$ W m$^{-2}$ are higher than the global mean values (Table S3). In terms of direct effects, in contrast, larger local changes were observed, as shown in Fig. 2D (shown here as a quantified sensitivity introduced below). This can be interpreted to emerge because, especially for times and locations with relatively high humidity (i.e., RH > 85%), the increase in hygroscopic

growth from increases in $\kappa_{\text{org}}$ and $\kappa_{\text{inorg}}$, leads directly to an increase in total aerosol-phase mass, thus increasing the aerosol optical depth (AOD) even in the absence of a change in mass scattering efficiency. In essence, Fig. 2 illustrates that the response in total light scattering, and therefore ari, to an increase in $\kappa$ is much more direct than the response of CCN concentrations, and thus aci, to the same factor. Figure 2D also shows some regions with positive values, but the pattern structure suggests that these are due to weather noise. Further note in this context that previous studies documented for organic compounds that the hygroscopicity measured above water saturation, as deduced from observed CCN activity, is higher than that observed in the subsaturated regime, as deduced from light scattering or hygroscopic tandem differential mobility analyser (HTDMA) data[43–45]. This means that the $\kappa_{\text{org}}$ derived here from CCN activity is likely an upper estimate when used for direct forcing calculations. However, while Table 1 suggests that the $\Delta\kappa$ between super- and subsaturated retrievals barely changes the global mean direct effects, it is still worth noting as it might affect conclusions on local effects as suggested by Fig. 2D.

We quantified the sensitivities $\xi(RF, \kappa)$ of the radiative forcing at the top of the atmosphere in relation to changes in $\kappa$ (i.e., the change in radiative forcing per unit change in $\kappa$) as follows:

$$\xi(RF, \kappa) = \frac{\Delta RF}{\Delta \kappa} \tag{5}$$

where RF can be either RFari or ERFaer and $\kappa$ can be either $\kappa_{\text{inorg}}$, $\kappa_{\text{org}}$ or $\kappa_{\text{org,inorg}}$ as the mass-weighted combination of $\kappa_{\text{org}}$ and $\kappa_{\text{inorg}}$. We obtained a globally averaged sensitivity of the RFari to changes in $\kappa_{\text{inorg}}$ of $\xi(\text{RFari}, \kappa_{\text{inorg}}) = -0.29 \pm 0.003$ W m$^{-2}$, which is about 20

times larger than the corresponding sensitivity to changes in $\kappa_{org}$ of $\xi(\mathrm{RFari}, \kappa_{org}) = -0.017 \pm 0.13\,\mathrm{W\,m^{-2}}$. The much higher $\xi(\mathrm{RFari}, \kappa_{inorg})$ compared to $\xi(\mathrm{RFari}, \kappa_{org})$ relates to the fact that also $\kappa_{inorg}$ is higher than $\kappa_{org}$, which entails a higher hygroscopic growth of inorganic than organic particles in the subsaturated humidity regime and, thus, a higher scattering cross section in aerosol–radiation interaction. When both $\kappa$ values are changed at the same time, we obtained $\xi(\mathrm{RFari}, \kappa_{org,inorg}) = -0.30 \pm 0.02\,\mathrm{W\,m^{-2}}$. Figure 2D shows that the local sensitivity can be substantially higher than the global average $\xi(\mathrm{RFari}, \kappa)$ values. For the overall aerosol-related effective radiative forcing, we obtained a sensitivity to changes in $\kappa_{inorg}$ of $\xi(\mathrm{ERFaer}, \kappa_{inorg}) = -0.47 \pm 0.01\,\mathrm{W\,m^{-2}}$, which is about 3 times larger than the corresponding sensitivity to changes in $\kappa_{org}$ of $\xi(\mathrm{ERFaer}, \kappa_{org}) = -0.15 \pm 0.62\,\mathrm{W\,m^{-2}}$. Note here the much larger uncertainty in $\xi(\mathrm{ERFaer}, \kappa_{org})$. For a change in both $\kappa$ values, we obtained $\xi(\mathrm{ERFaer}, \kappa_{org,inorg}) = -0.68 \pm 0.29\,\mathrm{W\,m^{-2}}$. As all these sensitivities are negative, an increase in aerosol hygroscopicity entails a decrease of the radiation at the top of the atmosphere and, therefore, an atmospheric cooling.

In essence, the aerosol hygroscopicity is of major importance for the atmospheric radiative budget. Our results here constrain and simplify $\kappa$ as an important prescribed parameter in global climate models. First, we provide a globally representative linear relationship of the additive hygroscopicity of organics and inorganic mass fractions, yielding $\kappa_{org} = 0.12 \pm 0.02$ and $\kappa_{inorg} = 0.63 \pm 0.01$ as recommended parameters for future model applications. Here, $\kappa_{org}$ should be used for both, primary and secondary organic aerosols. Second, the higher sensitivity of the model to changes in $\kappa_{inorg}$ relative to changes $\kappa_{org}$ stresses the importance of a correct representation of the abundance and hygroscopicity of the small number of relevant inorganic aerosol components (i.e., sulfates, nitrates, ammonium) in the models. In turn, this also implies that an experimental determination of the $\kappa$ of individual compounds within the thousands of different organics—with many of them being unidentified—is not essential to reduce uncertainties in the effective radiative forcing of aerosol particles. The variability of $\kappa$ within the organic aerosol fraction might be efficiently represented by average chemical properties such as the oxygen-to-carbon ratio[11,46]. More effort should be dedicated to correctly understand and represent the processes determining the abundance of the organics in the atmospheric multiphase systems and their representation in models, where the largest uncertainties are located in order to better predict the remaining important effects of particle organics as good as possible.

## Methods

### Aerosol mass spectrometric and size-resolved CCN data
The aerosol mass spectrometric (AMS) and size-resolved CCN data used in this study were obtained from the field studies listed below. AMS data on the non-refractory submicron aerosol fraction (i.e., mass concentrations of organics, nitrate, sulfate, ammonium, and chloride) were obtained either size-resolved from a quadrupole aerosol mass spectrometer (Q-AMS, Aerodyne Research Inc., Billerica, MA, USA), a high-resolution time-of-flight aerosol mass spectrometer (HR-ToF-AMS), or non-size-resolved from an aerosol chemical speciation monitor (ACSM, Aerodyne Research Inc.)[47,48]. Note that the AMS measurement of Cl⁻ is typically not quantitative due to a merely partial evaporation of the refractory NaCl as a major compound of sea salt aerosols.

CCN concentrations in the studies were measured with a continuous-flow streamwise thermal gradient CCN counter (CCNC, models CCN-100 or CCN-200, DMT, Longmont, CO, USA). In the instrument, the supersaturation ($S$) was cycled through different $S$ values, typically between 0.1 and 1.1%, defined by controlled temperature gradients inside the CCNC column. Particles with a critical supersaturation $S_c \geq S$ in the column are activated and form water droplets. Droplets with diameters larger 1 μm are detected by an optical particle counter (OPC) at the exit of the column. Further information on the CCN measurements and instrumentation can be found elsewhere[49,50]. Note that the CCN measurements were conducted in size-resolved mode, which is one established approach to retrieve $\kappa$, as outlined in detail elsewhere[16,19,51]. This applies to all three main data sets used here for the $\kappa_{org}$ and $\kappa_{inorg}$ retrieval (i.e., ATTO, PRIDE-PRD-2006, and CARE-Beijing-2006, shown in Fig. 1B) as well as for most of data sets from previous field studies outlined below.

### Field studies and data sets
This study integrates 16 field measurements worldwide, during which size-resolved CCN measurements and an aerosol mass spectrometer were operated. These studies can be broadly grouped into two categories: (i) three studies were selected for the retrieval of $\kappa_{inorg}$ and $\kappa_{org}$ according to criteria outlined below and (ii) 13 further data sets were implemented to evaluate to what extent the retrieved $\kappa_{inorg}$ and $\kappa_{org}$ values are representative worldwide. The original data sets of the following three studies represent the basis for the $\kappa_{inorg}$ and $\kappa_{org}$ retrieval:

ATTO, Brazil: The Amazon Tall Tower Observatory (ATTO) site is located in a remote area of the central Amazon rain forest (2.143° S, 59.000° W), -150 km northeast of the city of Manaus, Brazil. Previous studies provide detailed information on the ATTO site[52], the footprint area of its atmospheric observations[53], as well as the multi-year ACSM and size-resolved CCN measurements (March 2014 to February 2015)[9,19]. The conditions probed at ATTO ranged from pristine rain forest air to heavily polluted biomass burning smoke[54–56].

Guangzhou, China: The Program of Regional Integrated Experiments of Air Quality over the Pearl River Delta (PRIDE-PRD2006, short here PRD) in Guangzhou was conducted in July 2006 at a rural receptor site (23.550° N, 113.070° E) in the small village Backgarden -60 km northwest of the megacity Guangzhou on the outskirts of the densely populated center of the Pearl River Delta (PRD). Details on the campaign as well as CCN and AMS data sets can be found elsewhere[57–59].

Beijing, China: The Campaign of Air Quality Research in Beijing (CAREBeijing-2006, short here BEI) was conducted from 10 August to 09 September 2006 at a suburban site (39.510° N, 116.310° E) located on the campus of Huang Pu University in Yufa, -50 km south of the city center of Beijing (BEI). Details on the campaign as well as CCN and AMS datasets can be found elsewhere[60,61].

The following two data sets—which are published here for the first time—were included in the analysis as they add particularly relevant locations to the list of investigated environments:

Delhi, India: The Delhi-2018 campaign was performed to characterize the aerosol hygroscopicity in a strongly polluted region in India, comprehensive aerosol measurements during the Delhi-2018 campaign in the Indo-Gangetic plain were conducted[62,63]. The campaign site was on the campus of the India Meteorological Department (IMD, Delhi, India), which is located in the midst of heavy traffic and small industries. Instruments were placed in an air-conditioned container at -28 °C. The measurement period spanned from 05 Feb to 02 Mar 2018. This period marked the end of the winter in the Indo-Gangetic Plain and is characterized by light winds, mildly cold temperature, and high humidity[64].

Barbados: The measurements were conducted at the Barbados Atmospheric Chemistry Observatory (BACO, 13.165° N, −59.432° E) during th EUREC⁴A campaign. The measurement period spanned from 21 Jan to 22 Feb 2020 and a comprehensive campaign overview can be found in Stevens et al.[65]. Different aerosol populations were measured, spanning from clean marine background conditions to long-range transported aerosols, comprising dust as well as the episodic occurrence of mixtures of dust and black carbon[66].

The data sets from previously published studies outlined below were further merged into the present literature and data syntheses.

 

Our analysis either started from the original data (if available) or was based on digitized data points from relevant figures of the corresponding manuscripts, using the WebPlotDigitizer (https://automeris.io/WebPlotDigitizer/, last access 06 Feb 2022). This includes the following locations and measurements:

Puerto Rico: The Puerto Rico Aerosol Cloud Interaction Study (PRACS) took place in Puerto Rico from 9 to 18 Dec 2004. The relevant CCN and aerosol composition data has been reported in Allan et al.[22]. The site was dominated by north-easterly trade winds bringing clean, marine air masses, interrupted only by few episodes with moderate anthropogenic pollution. The aerosol probed during PRACS can be characterized as comprising mostly sea salt and non-sea salt sulfate. The amounts of organic matter in the aerosol were very low, which makes the data points from this study particularly relevant for the present analysis as the they represent the lowest $\epsilon_{org}$ levels in our data synthesis. For our analysis, the data points were digitized from figures in Allan et al.[22].

Taunus, Germany: The Feldberg Aerosol Characterization Experiment (FACE-2005) took place in June-July 2005 at the Taunus Observatory field site, which is a small mountain in a forested area. It represents a rural background site about 30–50 km of the Rhine-Main metropolitan area. The site receives diverse aerosol conditions, including urban plumes as well as aged rural air masses. The relevant CCN and aerosol composition data has been reported in Dusek et al.[24]. This data set can be regarded as characteristic for many other rural and/or semi-urban locations worldwide. For our analysis, the original data has been re-processed.

Jungfraujoch, Switzerland: The Cloud and Aerosol Characterization Experiment (CLACE-06) took place from 03 to 13 Mar 2007. The high-elevation research station is considered to be prevalently in the free troposphere, receiving aged background aerosol. The relevant CCN and aerosol composition data has been reported in Rose et al.[23]. For our analysis, the original data has been re-processed.

ZF2, Brazil: The Amazonian Aerosol Characterization Experiment (AMAZE-08) took place from 14 Feb to 12 Mar 2008 at the ZF2 site north of Manaus[67]. The relevant CCN and aerosol composition data has been reported in Gunthe et al.[16]. During the campaign, the site received clean rain forest air, interrupted by long-range transported plumes of African dust and biomass burning smoke. This data complements the ATTO data that was used for the $\kappa_{inorg}$ and $\kappa_{org}$ retrieval. For our analysis, the original data has been re-processed.

Hyytiälä, Finland: The SMEAR II station in the boreal forest in Hyytiälä, Finland is part of the European Integrated project on Aerosol Cloud Climate and Air Quality Interactions (EUCAARI) network and has recorded long term size-resolved CCN measurements. The data used belongs to the measurements from February 2009 to April 2012. The site is a rural background site that received clean and polluted air masses. The relevant CCN and aerosol composition data has been retrieved from a combination of papers[10,20,68].

Mexico (aircraft): The Megacities Impact on Regional and Global Environment (MIRAGE) Experiment took place over central Mexico in March 2006 in the course of research flights between 3 and 5 km altitude. MIRAGE was part of the Megacity Initiative: Local and Global Research Observations (MILAGRO) campaign[69]. The relevant CCN and aerosol composition data has been reported in Shinozuka et al.[25]. For our analysis, the data points were digitized from figures in Shinozuka et al.[25].

Mexico City, Mexico: The measurements outside of Mexico City were conducted from 16 to 31 Mar 2006 in the course of the MILAGRO and MIRAGE campaigns[69]. The relevant CCN and aerosol composition data has been reported in Lance et al.[70]. The measurements primarily probed "dense atmospheric pollutants trapped at high altitude, where rapid photochemical oxidation can take place", according to Lance et al.[70]. For our analysis, the data points were digitized from figures in Lance et al.[70]. Size-resolved $\epsilon_{org}$-$\kappa$ relationships are reported in Lance

et al.[70] for particle diameters of 40, 60, 80, and 100 nm. For comparability with the other studies, these $\epsilon_{org}$-$\kappa$ relationships were averaged in $\epsilon_{org}$ bins here.

USA (aircraft): The second part of the Intercontinental Chemical Transport Experiment−Phase B (INTEX-B) took place from 17 Apr to 15 May 2006 over the US West Coast in the course of research flights below 6 km altitude[71]. INTEX-B used the same instrumentation as during MILAGRO (see above). The relevant CCN and aerosol composition data has been reported in Shinozuka et al.[25]. The flights probed anthropogenic and biogenic emissions over extended areas as well as the influence of long-range transport from Asia. For our analysis, the data points were digitized from figures in Shinozuka et al.[25].

Taunus, Germany, shown in Fig. S4: The Ice Nuclei Research Unit− Taunus Observatory (INUIT-TO) campaign took place in August 2012 at the Taunus Observatory field site on Mt Kleiner Feldberg in a forested area in central Germany. The relevant CCN and aerosol composition data has been reported in Vogel et al.[38] (Fig. S4). During the campaign, fresh and aged aerosol populations were probed. Especially high fractions of sulfates covalently bonded to organic molecules are a focal point of this study as they cause substantially lower $\kappa$-values. Vogel et al.[38] argue that high fraction of organic acids, organosulfates, and nitrooxy-organosulfate affect the quantification of the organic and inorganic mass fractions by AMS. This limitation of standard AMS measurements can be solved by more sophisticated instrumentation on the chemical composition of aerosol particles and further systematic studies are required to resolve these effects.

SMEAR II, Finland, not shown here: The measurements were conducted from Mar to May 2007 and probed clean and polluted conditions. The relevant CCN and aerosol composition data has been reported in Cerully et al.[72] and the data points were digitized from the corresponding figures. This study was critically evaluated, though not included in our analysis because of a clear deviation from the overall linear relationship, as shown in Fig. S4. This is in line with the discussion in Cerully et al.[72], stating that especially "the $\kappa_{inorg}$ values are too low for typical inorganic compounds", likely due to a mismatch in the size ranges of the AMS and CCN counter measurements. The authors further conclude that "bulk measurements should be used with caution for representing characteristics of small mode particles."

Colorado, USA: In the course of the BEACHON (Bio-hydro-atmosphere Interactions of Energy, Aerosols, Carbon, $H_2O$, Organics, and Nitrogen) project in the semi-arid Manitou Experimental Forest, measurements were conducted from Mar 2010 to Feb 2011. The BEACHON site is representative of the Central Rocky Mountains montane zone and vegetation. It was chosen as it has minimal impact from nearby anthropogenic emissions and is located in a region with significant biogenic volatile organic compound (BVOC) emissions. Accordingly, the ambient submicron aerosol is expected to be dominated by organic compounds of mostly biogenic origin. The relevant CCN and aerosol composition data has been reported in Levin et al.[73] and[21], from where we digitized the data points for the present analysis.

For further details on the experimental conditions and measurements of aforementioned campaigns, refer to the original studies.

## Retrieval of $\kappa_{org}$ and $\kappa_{inorg}$

The following statistical procedure has been developed to obtain robust and widely representative results for $\kappa_{inorg}$ and $\kappa_{org}$:

Suitable data sets providing combined size-resolved CCN counter and AMS measurements in relevant environments were identified.

The statistically largest CCN and AMS data sets were used for the retrieval. Specifically, the number of $\epsilon_{org}$ vs $\kappa$ data points $n$ served as a criterion. Data sets with $n > 100$ were selected. This includes the ATTO ($n = 1503$), PRD ($n = 410$), and BEI ($n = 261$) data sets (see Table S2), while smaller data sets with $n < 100$ were omitted.

As shown by Gunthe et al.[61], the AMS and CCN counter measurements are most comparable for the largest particle diameters

measured at the lowest supersaturation (i.e., $S \sim 0.1\%$), which corresponds to accumulation mode particles serving as CCN. Note here that $\kappa$ is typically size-dependent and, therefore, focussing on one particular size range can introduce some bias. At the same time, $\kappa$ values are typically relatively constant across a given size mode[19,59]. Accordingly, the approach chosen here can be regarded as representative for accumulation mode particles as the prime source of CCN. This approach was also applied here. Note further that due to the AMS and ACSM detection limits, only data points with $f_{org} > 0.2$ were used[16]. The AMS time series were binned according to the time resolution of the CCN measurements, which yielded data points every 2.5–5 h, depending on the mode of operation.

For the three data sets with $n > 100$, the $\kappa$ data was binned onto an $\epsilon_{org}$ grid, with $\epsilon_{org}$ increments of 0.01, and the data were averaged within each bin (see Fig. S2). The binned data from ATTO, PRD, and BEI were merged, which yielded our optimized retrieval (see Fig. S1 and Fig. S2). The binning ensures that the data across the entire $\epsilon_{org}$ range is weighted equally and that the regression is not biased by differences in the density of the original $n$ data points.

A bivariate linear regression fit according to Cantrell[74] was applied, which takes experimental errors in $\epsilon_{org}$ and $\kappa$ into account.

Note that a binning and fitting of all data sets, compared to an averaging and fitting of the binned three largest data sets only (see above), did not yield a significantly different $\kappa_{org}$ and $\kappa_{inorg}$ retrieval within the range of the uncertainty. This shows that the optimized retrieval is statistically robust, independent of whether the smaller data sets are included or not. Binning all data sets with comparable counting error and averaging the binned results afterwards, however, seems to be the statistical most valid approach at this point.

### Retrieval of $M_{org}$ from $\kappa_{org}$
Following the approach in Gunthe et al.[16] and by interpreting $\kappa_{org}$ as an "effective Raoult parameter", we calculated an effective average molecular mass of the organic compounds ($M_{org}$) by adapting and applying the following equation from Rose et al.[51]

$$\kappa = i_s \frac{n_s V_w}{n_w V_s} = i_s \frac{\rho_s M_w}{\rho_w M_s} \quad \text{with} \quad i_s \approx \nu_s \Phi_s \qquad (6)$$

with $n$ as the numbers of moles, $\rho$ as the densities, and $M$ as molar masses of the dry solutes (s) and pure water (w) as well as the van't Hoff factor of the solute ($i_s$) with $\nu_s$ as the number of ions and $\Phi_s$ as the molar osmotic coefficient in aqueous solution, as follows

$$M_{org} \approx i_{org} \frac{\rho_{org} M_w}{\rho_w \kappa_{org}} \approx \nu_{org} \Phi_{org} \frac{\rho_{org} M_w}{\rho_w \kappa_{org}} \qquad (7)$$

with the $\tilde{i}_{org} \approx \nu_{org} \Phi_{org} \approx 1$, the molar density of water molecules in liquid water $\rho_w/M_w \approx 55 \text{ mol cm}^{-3}$, assuming the surface tension of pure water[75,76], and $\rho_{org} \approx 1.4 \text{ g cm}^{-3}$[77]. Note that $\kappa$ in Petters and Kreidenweis[7] has been defined based on the ratio of molar volumes. In the course of this study, we use the ratio of masses, which are routinely measured in aerosol studies.

### Maps and geographic information system (GIS) data
The analysis of geographic information system (GIS) data sets was conducted with the QGIS software package (version 3.12, QGIS development team, https://www.qgis.org/, last access 29 Jul 2023) using the coordinate reference of the World Geodetic System from 1984 (WGS84). The following GIS data sets have been used in this study:

Topography: The global topography data was obtained from Natural Earth (Free vector and raster map data, https://www.naturalearthdata.com, last access 29 Jul 2023)."

Coast lines and country borders: coast lines and country border data were obtained from Natural Earth (Free vector and raster map data, https://www.naturalearthdata.com, last access 29 Jul 2023).

Global water bodies: Maps of global water bodies were obtained from the European Space Agency (ESA) (https://www.esa-landcover-cci.org/?q=node/162, last access: 29 Jul 2023).

### ECHAM-HAM aerosol-climate model runs
We performed several model runs with the ECHAM–HAM model, version echam6.1–ham2.2–moz0.9[78]. The model is based on the ECHAM atmospheric general circulation model[39], the HAM interactive aerosol module[79,80], and the trace gas chemistry module MOZ[81] (the latter was disabled in these runs). Of most direct relevance to our study are the following parameterized processes contributing to warm-cloud–aerosol interactions: aerosol activation into cloud droplets according to Abdul-Razzak and Ghan[82]; diagnostic warm rain processes (autoconversion and accretion) according to Khairoutdinov and Kogan[83]; and aerosol scavenging according to Croft et al.[84,85]. We used the stratiform cloud scheme according to Lohmann and Roeckner[86], which was extended to double-moment microphysics by Lohmann et al.[87] and Lohmann and Hoose[88], with the Sundqvist et al.[89] critical-relative-humidity cloud cover parameterization. Thus, the model is capable, in principle, of representing both RFaci and rapid ACI adjustments. In terms of aerosol-radiation interactions, the scattering and absorption properties of the aerosol are computed in the model using Mie theory[80]. The scattering size parameter and volume-averaged refractive index are computed assuming an internally mixed aerosol, and taking into account the aerosol water content. Since the refractive index of aerosol water is very small—$2.0 \cdot 10^{-7}$ according to Zhang et al.[80]—and effects such as absorption enhancement in droplets is considered negligible, the effect of changes in $\kappa$ on absorption is expected to be very small compared to the impact on scattering. The optical properties used for the radiation computation are computed using all seven modes of the aerosol module. It is noted that the results are dependent on the predicted aerosol size distribution, as the wet radii change with the $\kappa$ value. Thus, the model is capable, in principle, of representing both RFari and RFaci, as well as rapid adjustments to both.

To reduce internal variability and to achieve low statistical uncertainty on the forcing components within a reasonable integration time, we used monthly varying but yearly repeating sea-surface temperature (SST) and sea ice cover (SIC) from the observed climatology and nudged the upper-level wind fields to the present-day ERA-Interim reanalysis[90] wind fields of the years 1980–2015. This reduces the internal variability without overconstraining the behavior of lower-tropospheric warm cloud and allow significant changes in global-mean forcing to emerge after a shorter integration time than would otherwise be required[91,92]. For the geographic distribution of CCN, 30-year runs (1985–2015) were required for robust patterns to emerge. Estimates of radiative forcing were computed by performing a pair of model runs with present-day SST, SIC, and wind fields, and aerosol (precursor) emissions estimates for either the year 2000 or the year 1850. Emissions were obtained from the AEROCOM-II ACCMIP dataset; anthropogenic emissions follow[93].

Model runs in various configurations were conducted: with the default Lin and Leaitch[94] or ref. Abdul-Razzak and Ghan[82] activation scheme, no or Petersik et al.[95] subgrid relative humidity variability for aerosol water uptake, and various $\kappa$ settings. We estimate RFari in each configuration following Ghan[96] using two runs, one with present-day (PD) emissions and one with preindustrial (PI) emissions, with a climatological annual cycle of SST and with large-scale wind fields nudged to ERA Interim reanalysis. The original model considers the following aerosol species with their compound-specific $\kappa$: sea salt ($\kappa_{SS} = 1.12$), mineral dust ($\kappa_{DU} = 0$), black carbon ($\kappa_{BC} = 0$), primary organic aerosol ($\kappa_{POA} = 0.06$), and

sulfate ($\kappa_{SU} = 0.60$)[7,80,97]. Note that the model uses sulfate (SU) as a surrogate species for all inorganic ions other than sea salt and dust. The overall $\kappa$ is then parameterized based on the additive mixing rule in Eq. (3). To take into account the effect of the optimized $\kappa_{inorg}$ and $\kappa_{org}$ on activation and thus RFaci as well as ERF, $\kappa$ had to be implemented in the Abdul-Razzak and Ghan[82] activation scheme.

We evaluated the effects of changing $\kappa$ of only the organic mass fraction, only the inorganic mass fraction, and both, the organic and inorganic mass fractions (see Table 1 and Table S3). The model's sensitivity was then assessed by means of the global mean direct radiative forcing (radiative forcing due to aerosol-radiation interactions, RFari) as well as effective radiative forcing due to the entire aerosol perturbation (ERFaer), including both aerosol-radiation and aerosol-cloud interactions. As expected, the changes in both $\kappa_{org}$ (from 0.06 to 0.12) and $\kappa_{inorg}$ (from 0.6 to 0.63) yield increases in magnitude of both RFari (owing to increased hygroscopic growth in the subsaturated regime) and ERFaer (which includes in addition an increase in CCN). The absolute change in ERFaer is only 0.2%, however the effect on the direct radiative effect is 10%. These changes are not very large, due to the fact that CCN concentrations are primarily determined by the aerosol number size distribution with particle chemical composition playing a secondary role[42]. For RFari, future measurements are needed to validate the retrieved values for subsaturated conditions.

The model is using the hygroscopicity parameter $B$ which was translated to a hygroscopicity parameter $\kappa$ as follows: From Petters and Kreidenweis[7], the saturation ratio $S$ can be expressed as a function of $\kappa$ in a modified Köhler equation:

$$S(D_s) = \frac{D_{wet}^3 - D_s^3}{D_{wet}^3 - (1-\kappa)D_s^3} \exp\left(\frac{4\sigma_{s/a}M_w}{RT\rho_w D_{wet}}\right). \tag{8}$$

For slightly supersaturated conditions with respect to water vapor saturation, as encountered in warm clouds, we can $-\ln S = \ln(1+s) \approx s$ (where $s$ is the supersaturation), so that Eq. (8) becomes

$$
\begin{aligned}
s(D) &\approx \ln\left(\frac{D_{wet}^3 - D_s^3}{D_{wet}^3 - (1-\kappa)D_s^3}\right) + \frac{4\sigma_{s/a}M_w}{RT\rho_w D_{wet}} \\
&= -\ln\left(\frac{D_{wet}^3 - (1-\kappa)D_s^3}{D_{wet}^3 - D_s^3}\right) + \frac{4\sigma_{s/a}M_w}{RT\rho_w D_{wet}} \\
&= -\ln\left(1 + \frac{\kappa D_s^3}{D_{wet}^3 - D_s^3}\right) + \frac{4\sigma_{s/a}M_w}{RT\rho_w D_{wet}} \\
&\approx -\frac{\kappa D_s^3}{D_{wet}^3 - D_s^3} + \frac{4\sigma_{s/a}M_w}{RT\rho_w D_{wet}} \\
&\approx -\underbrace{\kappa \frac{D_s^3}{D_{wet}^3}}_{BD_s^3/D_{wet}^3} + \underbrace{\frac{4\sigma_{s/a}M_w}{RT\rho_w D_{wet}}}_{A/D_{wet}},
\end{aligned}
\tag{9}
$$

with the first term being the solute term, describing the Raoult effect, and the second term the curvature term for the Kelvin effect. Under conditions for which the $-s \ll 1$ and $D_s^3 \ll D_{wet}^3$ hold, $B = \kappa$.

## Data availability
Most data sets on cloud condensation nuclei concentrations and aerosol chemical composition used in this study have been presented in previous studies and are available through the original publications as cited in the manuscript. Data from the figures of this study are available under https://edmond.mpdl.mpg.de/ via https://doi.org/10.17617/3.HG0GHF (Pöhlker et al., 2023).

## Code availability
The ECHAM6-HAM model is made available to the scientific community under https://redmine.hammoz.ethz.ch/ (last access: 09 July 2023). The availability is regulated under the Software Licence

Agreement that can be downloaded from https://redmine.hammoz.ethz.ch/attachments/download/291/License_ECHAM-HAMMOZ_June2012.pdf (last access: 09 July 2023).

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

## Acknowledgements
We thank Diana Rose, Amit Sharma, James Allan, Upasana Panda, Joseph Prospero, Johannes Schneider, and Bjorn Stevens for support and

fruitful discussions. This work used resources of the Deutsches Klimar-echenzentrum (DKRZ) granted by its Scientific Steering Committee (WLA) under project ID bb1036. M.L.P., C.P., A.P., O.O.K., D.W. and U.P. acknowledge funding by the Max Planck Society (MPG). C.P. and U.P. acknowledges funding from the Bundesministerium für Bildung und Forschung (BMBF contracts 01LB1001A, 01LK1602B, and 01LK2101B) in the context of the ATTO project. J.Q. acknowledges funding from the European Union's Horizon 2020 research and innovation programme under grant agreement No 821205 (FORCES). B.E. acknowledges fund-ing from the French National Research Agency (grant no. ANR-17-MPGA-0013). This work was supported by NERC and the Newton Fund through the PROMOTE project of the APHH-Delhi programme, grant ref. NE/P016480/1. The ECHAM-HAMMOZ model is developed by a consortium composed of ETH Zurich, Max Planck Institut für Meteorologie, For-schungszentrum Jülich, University of Oxford, the Finnish Meteorological Institute and the Leibniz Institute for Tropospheric Research, and man-aged by the Leibniz Institute for Tropospheric Research (TROPOS).

## Author contributions

M.L.P. and U.P. conceptualized the study. D.W. was responsible for the data management and curation. M.L.P., D.W. and C.P. conducted the data analysis. M.L.P., C.P., H.C., P.G., C.J.G., S.S.G., O.O.K., G.M., S.S.R., E.R.V. and H.M.R. conducted the field observations in the course of the Delhi-2018 and EUREC$^4$A campaigns. M.L.P., D.W., J.Q., J.M. and K.B. wrote the software and programs for the data analysis and modelling. A.P., B.E., S.H., L.P., Y.W., H.H., P.A. and M.O.A .validated the results and conclusions of the study. M.L.P. and C.P. created the figures and tables. M.L.P. and C.P. wrote the manuscript. All authors contributed to the scientific discussion and interpretation of the results as well as to the final paper writing.

## Funding

## Competing interests

The authors declare no competing interests.
