## [Peer Review File · Nature Communications]

REVIEWER COMMENTS

Reviewer #1 (Remarks to the Author):

The authors have used data from a number of field campaigns spanning a wide variety of site types (urban, forested, coastal, among others). They show that observed overall hygroscopicities can be recovered via a simple 2-component linear mixing rule, combining an averaged inorganic $k_{\text{inorg}} = 0.63$ with an averaged organic $k_{\text{org}} = 0.12$, weighted by their mass fractions. Further, they have conducted estimates of the perturbations to direct and indirect radiative forcings, modifying the assumed component hygroscopicities. The authors conclude that forcing predictions, on a global scale, are not very sensitive to the exact choice of organic component hygroscopicity, and suggest this finding can be used to focus ongoing research toward more sensitive factors in order to reduce remaining uncertainties.

The paper is interesting and makes good use of data from past studies to present them in a new way. I think there will be interest in this topic, especially from those outside the aerosol-cloud interaction community. The paper also aims to provide specific recommendations for future directions, and these will likely generate numerous additional citations.

I recommend that this manuscript be accepted for publication, but suggest that the authors modify / qualify their helpful recommendations, as I explain below in Major Comments. I also offer some other comments for their consideration.

Major Comments

The paper argues that “the aerosol hygroscopicity is of major importance for the atmospheric radiative budget.” This is certainly true for direct radiative effects, which are sensitive to relative humidity (RH) and to the representation of the response of aerosol water contents to changing RH. It is also argued, quite reasonably, that further refining the precise hygroscopicity of the complex mixtures of organic species typically comprising the atmospheric aerosol is less important than other factors. The final statement is:

“Instead, more effort should be dedicated to correctly understand and represent the processes determining the abundance of the organics in the atmospheric multiphase systems and their representation in models, where the largest uncertainties are located in order to better predict the remaining important effects of particle organics [on radiative forcing] as good as possible.”

With respect to the effects of composition on CCN concentrations, and hence on the simulated indirect forcings, the statement of the importance of the aerosol hygroscopicity seems to contradict the last line of the first paragraph:

“The sensitivity of global climate forcing to changes in k_{org} and k_{inorg} is small, which constrains a critically important aspect of global climate modelling.”

To avoid possible confusion, I suggest that differences between direct and indirect forcing calculations be made more explicit, as follows.

For indirect forcing / CCN concentration predictions: I agree with the observation that overall, the sensitivity to aerosol composition (outside of extremes) should be small. Of the three factors (other than temperature) that control the CCN spectrum, namely total number concentration, particle size, and particle hygroscopicity, the sensitivity to number is easily the largest. Further, the simplified Kohler relationship between critical supersaturation and aerosol properties shows that the critical supersaturation varies with particle size to the $-3/2$ power, but hygroscopicity only to the $-1/2$ power. The authors also note these factors in the methods, on page 14, lines 24-25, citing Dusek et al. (2006). Hence the exact value of hygroscopicity is generally the least important determinant of CCN activity, and the results in Figure 2B, which show close to zero changes in simulated CCN active at 0.3% supersaturation, are not surprising. I think it would be helpful to make the total sensitivities clear and to include these other factors when discussing the recommendations for future work to constrain indirect forcing.

The above discussion clarifies why, in the simulations, the results shown in Figure 2C indicate that in all locations the sensitivity of the indirect forcing to changes in hygroscopicity is small. This is because the CCN spectrum that is input to the model at each grid point does not change that much when only hygroscopicity is modified over a relatively small range, if all other more sensitive factors (total number concentrations and aerosol size distributions) are the same.

For direct forcing: In contrast, there are larger local changes in direct forcing due to hygroscopicity changes, as discussed for Figure 2D. I interpret this finding to emerge because, especially for times and locations with relatively high humidity ($RH > 85\%$), the increase in hygroscopic growth from increasing the hygroscopicity of both components, leads directly to an increase in total aerosol-phase mass, thus increasing AOD even in the absence of a change in mass scattering efficiency. In other words: the response in total light scattering (or AOD) to an increase in hygroscopicity is much more direct, than is the response of CCN concentrations to the same factor.

It's not entirely clear how the model handles the direct forcing calculation; I assume that a new mass scattering efficiency is also selected due to the change in wet particle diameter with a more hygroscopic aerosol. However, as long as the size distribution initially peaks near its most-effective diameters, the variation in mass scattering efficiency should be relatively small. Regions in Figure 2D with positive sensitivities are therefore somewhat curious (do those contradict the statement on page 2, line 8?) Any dependence on mass scattering efficiency will highlight the importance of simulating the aerosol size distribution accurately for accurate direct forcing estimates. As for the most important factors to resolve for direct forcing, I would strongly concur with the concluding statement recommending attention to the accurate simulation of total aerosol-phase mass, as it directly multiplies the mass scattering efficiency.

If the authors agree with the above interpretations of Figure 2, I believe it would be helpful to provide some of these additional interpretations in the main text, and to write separate conclusions and recommendations for the direct and indirect forcings.

Other comments

1. The title refers to "radiative forcing" in general but the introduction and data focus on CCN observations. As noted above, calculations are presented for both the direct (subsaturated) and indirect (CCN-related) forcings. I think it would be helpful to distinguish the direct and indirect cases more clearly. Also, I think only contributions from non-absorbing species are considered here. There remain significant uncertainties in estimating aerosol absorption and its role in radiative forcing, so it might be prudent to mention that here.

2. Page 2, line 8: Particles of the same size but higher hygroscopicity act more readily as CCN (need to point out role of size here).

3. Page 2, line 26: for work on resolving the value of k_{org} based on organic compound characteristics, please include this reference:

Petters, SS, D Pagonis, MS Claflin, EJT Levin, MD Petters, PJ Ziemann, SM Kreidenweis, Hygroscopicity of organic compounds as a function of carbon chain length, carboxyl, hydroperoxy, and carbonyl functional groups, *J. Phys. Chem. A*, 2017, 121(27), 5164-5174; doi 10.1021/acs.jpca.7b04114.

4. Page 3, equation 4: This assumes that the inorganic component is composed solely of hygroscopic salts. There are often mineral components and elemental carbon, which are not detected by the AMS. Therefore this equation is an approximation, and in general, the presence of non- or weakly-hygroscopic

compounds as minor constituents is probably subsumed into the deduced value of k_{org} and/or k_{inorg} . This equation merits some additional explanation of the assumptions inherent here, including the upper limit on aerosol size that is imposed by the AMS, and the next point below.

5. As noted in several places in the manuscript, the use of mass fraction instead of volume fraction to weight water contents is an approximation, compared with the ZSR rule that is typically invoked. It is argued that the difference is small. For the densities assumed here, that is true for high mass fractions of organics ($e_{\text{org}} > 0.75$). For lower mass fractions, down to their limit of $e_{\text{org}} > 0.2$, the differences can be as large as 15% with the volume fractions larger than the corresponding mass fraction, since the organic is less dense. Using mass fraction as the independent variable for a linear fit has, I believe, the effect of predicting slightly lower k_{inorg} (possibly suggested in Figure 1C?), which may partially compensate for nonhygroscopic components that are not explicitly included in the mixing rule.

6. Page 3, lines 22-26: the authors pose two questions. Question 1 could be phrased more explicitly, that is, “Is there a pair of robust k_{org} and k_{inorg} values that is globally representative of aerosol water contents, above and below water saturation?” In Question 2, does “radiative forcing” only mean direct radiative forcing? If both direct and indirect, please specify more clearly.

7. Bottom of page 3, and comprising the description of how k was retrieved, as well as Methods on page 9, lines 10-15: It seems like there is something missing in the description that may make this misleading to those from other communities. It is stated that AMS and CCN counter measurements were needed for the regression (e.g., page 9, line 17; page 12, line 24). However, neither of these is a measurement of k ; additional data or methodologies are needed, most likely size distributions that can be used to derive effective k s, or a size-resolved CCN measurement, which I believe is what was used here. Gunthe et al (2011), cited herein, use size-resolved CCN spectra to obtain paired supersaturations and activation diameters, from which effective hygroscopicities could be deduced. It is also possible to measure total CCN at a particular supersaturation and also the total aerosol size distribution, assume that the largest particles activate first, and estimate an activation diameter in this way in order to compute the effective hygroscopicity. In the methods on page 9, the operation of the CCN counter is not as a size-resolved measurement. If the Gunthe et al. approach was used for all the data sets, then please make this clear in the description on page 9, and also note that this requires an experimental setup that is different than simply operating a CCN counter to obtain total activated particles as a function of supersaturation, which is implied here.

8. Page 5, line 15 and following: It is argued why the method works even though dust and sea salt are not represented in the AMS data. This is likely a result of the upper size limit of the AMS, since it has been shown that sea spray composition varies strongly with size. This size dependence is implied with the statement “accounts for most of the CCN”. Further, Table S2 explains the sizes that the data are specific to, which also limits the expected composition.

9. Page 6, lines 4-5: Note that for organic compounds, it has frequently been documented that the hygroscopicity measured above water saturation – that is, deduced from observed CCN activity – is higher than that observed in the subsaturated regime, i.e., deduced from light scattering or HTDMA data. The authors should note this as a caveat to their methodology, in that the derived k_{org} is likely an upper estimate when used for direct forcing calculations. As seen in Table 1, the global mean value for direct effects is barely changed even with the higher k_{org} , but this may affect conclusions around local effects (Figure 2D). The authors could perhaps show, in the Supplement, the equivalents of Figure 2D for the second and third cases in Table 1 to address this possibility.

10. Page 8, lines 12-14: the hygroscopicity of the inorganic components (when measured ions are interpreted as various salts, but excluding sodium compounds which are also excluded herein) does not vary that much, which is also stated on page 3, line 20.

11. Just an observation: For a nonabsorbing aerosol, as assumed here, simulation of the size distribution can be an important factor, in order to select an appropriate mass scattering efficiency. Composition also plays a significant role, of course. Interestingly, prior work showed that the ZSR mixing rule (the basis for the kappa mixing rule) also applies to scattering calculations and the efficiency curves could be weighted by the composition. Also interestingly, depending on how the calculation is set up, there is less dependence on the degree of neutralization of sulfate than might be expected. This is because the lower hygroscopicity of fully neutralized sulfate is balanced by the additional ammonium mass, while the lower ammonium mass per sulfate ion in bisulfate is compensated by the higher water content; variation in refractive index between those cases is a minor factor.

12. If you would like to add a North American continental site, the following publication also used AMS and size-resolved CCN data in a two-component model to draw a similar conclusion ($k_{org} = 0.13$, assuming $k_{inorg}=0.6$):

Levin, E. J. T., Prenni, A. J., Palm, B. B., Day, D. A., Campuzano-Jost, P., Winkler, P. M., Kreidenweis, S. M., DeMott, P. J., Jimenez, J. L., and Smith, J. N.: Size-resolved aerosol composition and its link to hygroscopicity at a forested site in Colorado, *Atmos. Chem. Phys.*, 14, 2657–2667, <https://doi.org/10.5194/acp-14-2657-2014>, 2014.

Review prepared by Sonia Kreidenweis

Reviewer #2 (Remarks to the Author):

Please see the attached Word document for the detailed review comments.

Responses to the review of the manuscript “Global organic and inorganic aerosol hygroscopicity and its effect on radiative forcing”, by Pöhlker et al., submitted for publication in Nature Communications

Dear editor, we would like to thank you and both referees for the valuable comments and useful suggestions to improve our manuscript. Below, you will find our responses to the individual comments of the referees along with a description of changes that we did in the manuscript. We have used the following color code:

- **Black:** the referee’s comments.
- **Blue:** the authors’ responses.
- **Red:** Quotes from the original manuscript.
- **Red & italic:** Text modifications in the manuscript related to the referee comments.

Responses to Referee #1 (Sonia Kreidenweis)

The authors have used data from a number of field campaigns spanning a wide variety of site types (urban, forested, coastal, among others). They show that observed overall hygroscopicities can be recovered via a simple 2-component linear mixing rule, combining an averaged inorganic $k_{inorg} = 0.63$ with an averaged organic $k_{org} = 0.12$, weighted by their mass fractions. Further, they have conducted estimates of the perturbations to direct and indirect radiative forcings, modifying the assumed component hygroscopicities. The authors conclude that forcing predictions, on a global scale, are not very sensitive to the exact choice of organic component hygroscopicity, and suggest this finding can be used to focus ongoing research toward more sensitive factors in order to reduce remaining uncertainties.

The paper is interesting and makes good use of data from past studies to present them in a new way. I think there will be interest in this topic, especially from those outside the aerosol-cloud interaction community. The paper also aims to provide specific recommendations for future directions, and these will likely generate numerous additional citations.

I recommend that this manuscript be accepted for publication, but suggest that the authors modify / qualify their helpful recommendations, as I explain below in Major Comments. I also offer some other comments for their consideration.

We appreciate the positive evaluation and the valuable comments by referee #1. The specific comments are addressed below in detail.

Major Comments

[R1.1] The paper argues that “the aerosol hygroscopicity is of major importance for the atmospheric radiative budget.” This is certainly true for direct radiative effects, which are sensitive to relative humidity (RH) and to the representation of the response of aerosol water contents to changing RH. It is also argued, quite reasonably, that further refining the precise hygroscopicity of the complex mixtures of organic species typically comprising the atmospheric aerosol is less important than other factors. The final statement is:

“Instead, more effort should be dedicated to correctly understand and represent the processes determining the abundance of the organics in the atmospheric multiphase systems and their representation in models, where the largest uncertainties are located in order to better predict the remaining important effects of particle organics [on radiative forcing] as good as possible.”

With respect to the effects of composition on CCN concentrations, and hence on the simulated indirect forcings, the statement of the importance of the aerosol hygroscopicity seems to contradict the last line of the first paragraph:

“The sensitivity of global climate forcing to changes in k_{org} and k_{inorg} is small, which constrains a critically important aspect of global climate modelling.”

[A1.1a] We improved the last statement of the abstract as shown below and hope that this clarifies this issue.

“By showing that the sensitivity of global climate forcing to changes in k_{org} and k_{inorg} is small, we constrain a critically important aspect of global climate modelling.”

To avoid possible confusion, I suggest that differences between direct and indirect forcing calculations be made more explicit, as follows.

For indirect forcing / CCN concentration predictions: I agree with the observation that overall, the sensitivity to aerosol composition (outside of extremes) should be small. Of the three factors (other than temperature) that control the CCN spectrum, namely total number concentration, particle size, and particle hygroscopicity, the sensitivity to number is easily the largest. Further, the simplified Köhler relationship between critical supersaturation and aerosol properties shows that the critical supersaturation varies with particle size to the $-3/2$ power, but hygroscopicity only to the $-1/2$ power. The authors also note these factors in the methods, on page 14, lines 24-25, citing Dusek et al. (2006). Hence the exact value of hygroscopicity is generally the least important determinant of CCN activity, and the results in Figure 2B, which show close to zero changes in simulated CCN active at 0.3% supersaturation, are not surprising. I think it would be helpful to make the total sensitivities clear and to include these other factors when discussing the recommendations for future work to constrain indirect forcing.

The above discussion clarifies why, in the simulations, the results shown in Figure 2C indicate that in all locations the sensitivity of the indirect forcing to changes in hygroscopicity is small. This is because the CCN spectrum that is input to the model at each grid point does not change that much when only hygroscopicity is modified over a relatively small range, if all other more sensitive factors (total number concentrations and aerosol size distributions) are the same.

For direct forcing: In contrast, there are larger local changes in direct forcing due to hygroscopicity changes, as discussed for Figure 2D. I interpret this finding to emerge because, especially for times and locations with relatively high humidity ($RH > 85\%$), the increase in hygroscopic growth from increasing the hygroscopicity of both components, leads directly to an increase in total aerosol-phase mass, thus increasing AOD even in the absence of a change in mass scattering efficiency. In other words: the response in total light scattering (or AOD) to an increase in hygroscopicity is much more direct, than is the response of CCN concentrations to the same factor.

[A1.1b] Thanks for these helpful comments and valuable suggestions. We have modified the entire paragraph on page 6 to make the differences between direct and indirect forcing more explicit. We have implemented the thoughts and arguments as follows:

“In principle, the aerosol effects through ari and aci could be treated approximately additively, which allows to retrieve RF_{aci} as $ERF_{aer} - RF_{ari}$ from Table 1. We refrain from reporting these values here, however, as this approach does not account for the various adjustments in the system upon ari or aci changes and also cannot be obtained directly from the model. In terms of aci, the replacement of κ yields the geographic distributions of the changes in CCN concentrations (ΔN_{CCN}) and in the effective radiative forcing (ΔERF_{SW}) shown in Fig. 2B and C. These changes in ΔN_{CCN} and ΔERF_{SW} are comparatively small, which is not surprising since out of the aerosol-related factors that control the CCN spectrum – which are total number concentration, particle size, and particle hygroscopicity – the sensitivity to number is the largest. Moreover, the simplified Köhler equation

(Eq. 9) underlines that the exact value of κ is generally the least important determinant of CCN activity as S varies with D to the power of three and just linearly with κ . The largest ΔNCCN and ΔERFSW values observed in Fig. 2B and C were found in regions with a high organic aerosol burden and corresponding values (filtered for high OC) of $\Delta\text{RFari}=-0.049\text{Wm}^{-2}$ and $\Delta\text{ERFaer}=-0.149\text{Wm}^{-2}$ are higher than the global mean values (Table S4). In terms of ari, in contrast, larger local changes were observed as shown in Fig. 2D (shown here as a quantified sensitivity introduced below). This can be interpreted to emerge because, especially for times and locations with relatively high humidity (i.e., $\text{RH} > 85\%$), the increase in hygroscopic growth from increases in k_{org} and k_{inorg} , leads directly to an increase in total aerosol-phase mass, thus increasing the aerosol optical depth (AOD) even in the absence of a change in mass scattering efficiency. In essence, Fig. 2 illustrates that the response in total light scattering, and therefore ari, to an increase in κ is much more direct, than the response of CCN concentrations, and aci, to the same factor.

[R1.2] It's not entirely clear how the model handles the direct forcing calculation; I assume that a new mass scattering efficiency is also selected due to the change in wet particle diameter with a more hygroscopic aerosol. However, as long as the size distribution initially peaks near its most-effective diameters, the variation in mass scattering efficiency should be relatively small. Regions in Figure 2D with positive sensitivities are therefore somewhat curious (do those contradict the statement on page 2, line 8? *“First, more hygroscopic aerosol particles grow to larger diameters at relative humidity <100 %, leading to a greater scattering cross section and thus stronger radiative forcing by aerosol-radiation interactions (RFari).”*). Any dependence on mass scattering efficiency will highlight the importance of simulating the aerosol size distribution accurately for accurate direct forcing estimates. As for the most important factors to resolve for direct forcing, I would strongly concur with the concluding statement recommending attention to the accurate simulation of total aerosol-phase mass, as it directly multiplies the mass scattering efficiency. If the authors agree with the above interpretations of Figure 2, I believe it would be helpful to provide some of these additional interpretations in the main text, and to write separate conclusions and recommendations for the direct and indirect forcings.

[A1.2] The way the radiative effect of aerosols is computed in the model indeed has been described too superficially. We added the relevant information in the supplemental section "ECHAM-HAM aerosol-climate model runs" for better clarification as follows:

“In terms of aerosol-radiation interactions, the scattering and absorption properties of the aerosol are computed in the model using Mie theory (Zhang et al., 2012). The scattering size parameter and volume-averaged refractive index are computed assuming internally mixed aerosol, and taking into account the aerosol water content. With this, both aspects of aerosol-radiation interactions may be affected. The optical properties used for the radiation computation are computed using all seven modes of the aerosol module. It is noted that the results are dependent on the predicted aerosol size distribution, as the wet radii change with the k value. Thus, the model is capable, in principle, of representing both RFari and RFaci, as well as rapid adjustments to both.”

So the reviewer is completely right, that the mass scattering efficiency changes, since the aerosol water content is altered. As for the regions with positive values in Fig. 2D, we believe these to be due to weather noise, rather than to be a clear signal. The following sentence on this has been inserted into the revised manuscript on page 8:

“Figure 2D also shows some regions with positive values, but the pattern structure suggests that these are due to weather noise.”

Other comments

[R1.3] The title refers to “radiative forcing” in general but the introduction and data focus on CCN observations. As noted above, calculations are presented for both the direct (subsaturated) and indirect (CCN-related) forcings. I think it would be helpful to distinguish the direct and indirect cases more clearly. Also, I think only contributions from non-absorbing species are considered here. There remain significant uncertainties in estimating aerosol absorption and its role in radiative forcing, so it might be prudent to mention that here.

[A1.3] The reviewer is right, we look into both, direct and indirect radiative forcings. However, with our method it is only possible to separate direct radiative forcing, and total effective radiative forcing. The indirect radiative forcing cannot be singled out. One might assume that the semi-direct effect (adjustments to aerosol-radiation interactions in the IPCC nomenclature) is small, and so assume the total ERF minus the RF_{dir} is the ERF_{ind}. But since we report all numbers we can obtain, we think we did as best as we could.

We consider both absorbing and non-absorbing species, but the impact of altered aerosol water uptake on the scattering is much larger than the one on absorption. This is now clarified in the model description on page 15 in the supplement through the following statement:

“Since the refractive index of aerosol water is very small - $2.0 \cdot 10^{-7}$ according to Zhang et al., 2012 - and effects such as absorption enhancement in droplets is considered negligible, the effect of changes in κ on absorption is expected to be very small compared to the impact on scattering.”

[R1.4] Page 2, line 8: Particles of the same size but higher hygroscopicity act more readily as CCN (need to point out role of size here).

[A1.4] Thanks. We modified the sentence as suggested to avoid confusion. It reads now as follow:

“Second, particles of the same size but higher hygroscopicity act more readily as cloud condensation nuclei (CCN), which activate into cloud droplets, leading to a larger cloud droplet number, longer-lived clouds and stronger forcing by aerosol--cloud interactions (RF_{ind}), and, together with the cloud adjustments, this entails a stronger ERF_{tot}.”

[R1.5] Page 2, line 26: for work on resolving the value of k_{org} based on organic compound characteristics, please include this reference:

Petters, SS, D Pagonis, MS Clafin, EJT Levin, MD Petters, PJ Ziemann, SM Kreidenweis, Hygroscopicity of organic compounds as a function of carbon chain length, carboxyl, hydroperoxy, and carbonyl functional groups, J. Phys. Chem. A, 2017, 121(27), 5164-5174; doi 10.1021/acs.jpca.7b04114.

[A1.5] We have included the reference as suggested.

[R1.6] Page 3, equation 4: This assumes that the inorganic component is composed solely of hygroscopic salts. There are often mineral components and elemental carbon, which are not detected by the AMS. Therefore this equation is an approximation, and in general, the presence of non- or weakly-hygroscopic compounds as minor constituents is probably subsumed into the deduced value of k_{org} and/or k_{inorg} . This equation merits some additional explanation of the assumptions inherent here, including the upper limit on aerosol size that is imposed by the AMS, and the next point below.

[A1.6a] We have modified the corresponding statements as suggested to make the inherent assumptions transparent by adding the following statement:

“Note that Eq. 4 is an approximation and assumes that the inorganic component is composed solely of hygroscopic salts. As mineral components and elemental carbon are not detected by the AMS, the presence of non- or weakly-hygroscopic compounds as minor constituents is subsumed into the deduced value of k_{org} and/or k_{inorg} . The upper limit of AMS measurements (i.e., 50 % transmission) and therefore the data used in Eq. 4 is typically ~600 nm (Liu et al., 2007). Further, Eq. 3 is based on volume fractions, whereas Eq. 4 uses mass fractions instead, which is an approximation. It is justified here as the densities, ρ , are sufficiently similar (i.e., $\rho_{(NH_4)_2SO_4} = 1.77 \text{ g cm}^{-3}$, $\rho_{NH_4NO_3} = 1.72 \text{ g cm}^{-3}$, $\rho_{org,average} = 1.4 \text{ g cm}^{-3}$ (17)), especially for high ϵ_{org} (i.e., > 0.75). For lower ϵ_{org} (i.e., ≈ 0.25), however, the difference between volume and mass fractions can be up to 15 % (with volume fractions being larger), which has to be considered in the interpretation of the results.”

Page 5 lines 29-30 the authors write, “the assumption of an internally mixed aerosol as an average of the global composition is justified (31).”

[A1.6b] Please refer to [R2.4] and [A2.4], where the aspect of the aerosol mixing state has been addressed in detail.

[R1.7] As noted in several places in the manuscript, the use of mass fraction instead of volume fraction to weight water contents is an approximation, compared with the ZSR rule that is typically invoked. It is argued that the difference is small. For the densities assumed here, that is true for high mass fractions of organics ($\epsilon_{org} > 0.75$). For lower mass fractions, down to their limit of $\epsilon_{org} > 0.2$, the differences can be as large as 15% with the volume fractions larger than the corresponding mass fraction, since the organic is less dense. Using mass fraction as the independent variable for a linear fit has, I believe, the effect of predicting slightly lower k_{inorg} (possibly suggested in Figure 1C?), which may partially compensate for nonhygroscopic components that are not explicitly included in the mixing rule.

[A1.7] We appreciate this comment and implemented this thought on page 3, line 8ff, as outlined in the course of our response [A1.6].

[R1.8] Page 3, lines 22-26: the authors pose two questions. Question 1 could be phrased more explicitly, that is, “Is there a pair of robust k_{org} and k_{inorg} values that is globally representative of aerosol water contents, above and below water saturation?” In Question 2, does “radiative forcing” only mean direct radiative forcing? If both direct and indirect, please specify more clearly.

[A1.8] We thank the referee for the good advice.

We have changed question 1 from “Is there a pair of robust κ_{org} and κ_{inorg} values that is globally representative and, thus, a suitable choice for general application in aerosol–climate models?” to “Is there a pair of robust κ_{org} and κ_{inorg} values that is globally representative of aerosol water contents, above and below water saturation and, thus, a suitable choice for general application in aerosol–climate models?”.

The second question was changed from “How sensitive are the model predictions in terms of CCN concentrations and radiative forcing to the choice of robust κ_{org} and κ_{inorg} ?” to “How sensitive are the model predictions in terms of CCN concentrations as well as direct and indirect radiative forcing to the choice of robust κ_{org} and κ_{inorg} ?”.

[R1.9] Bottom of page 3, and comprising the description of how k was retrieved, as well as Methods on page 9, lines 10-15: It seems like there is something missing in the description that may make this misleading to those from other communities. It is stated that AMS and CCN counter measurements were needed for the regression (e.g., page 9, line 17; page 12, line 24). However, neither of these is a measurement of k ; additional data or methodologies are needed, most likely size distributions that

can be used to derive effective κ s, or a size-resolved CCN measurement, which I believe is what was used here. Gunthe et al (2011), cited herein, use size-resolved CCN spectra to obtain paired supersaturations and activation diameters, from which effective hygroscopicities could be deduced. It is also possible to measure total CCN at a particular supersaturation and also the total aerosol size distribution, assume that the largest particles activate first, and estimate an activation diameter in this way in order to compute the effective hygroscopicity. In the methods on page 9, the operation of the CCN counter is not as a size-resolved measurement. If the Gunthe et al. approach was used for all the data sets, then please make this clear in the description on page 9, and also note that this requires an experimental setup that is different than simply operating a CCN counter to obtain total activated particles as a function of supersaturation, which is implied here.

[A1.9] We agree – the description is incomplete and misleading here. We modified that statement on page 3:

“First, we conducted a systematic retrieval of κ by merging extensive data sets from AMS and CCN counters obtained in contrasting environments (Fig. 1A) to calculate a pair of broadly representative κ_{org} and κ_{inorg} values.”

as follows

“First, we conducted a systematic retrieval of κ by merging extensive data sets from AMS and size-resolved CCN measurements obtained in contrasting environments (Fig. 1A) to calculate a pair of broadly representative κ_{org} and κ_{inorg} values.”

On page 9, lines 10 to 15, it is actually mentioned that the CCN measurements were conducted in size-resolved mode. Nevertheless we modified the text to emphasize this aspect even more, as we agree with the referee that this should be concisely stated. The original text on page 5:

“The aerosol mass spectrometric (AMS) and size-resolved CCN data used in this study were obtained from the field studies listed below. [...]. CCN concentrations in the studies were measured with a continuous-flow streamwise thermal gradient CCN counter (CCNC, models CCN-100 or CCN-200, DMT, Longmont, CO, USA). In the instrument, the supersaturation (S) was cycled through different S values, typically between 0.1 and 1.1 %, defined by controlled temperature gradients inside the CCNC column. Particles with a critical supersaturation $S_c > S$ in the column are activated and form water droplets. Droplets with diameters larger 1 μm are detected by an optical particle counter (OPC) at the exit of the column. Further information on the CCN measurements and instrumentation can be found elsewhere (refs).”

was modified as follows

“The aerosol mass spectrometric (AMS) and size-resolved CCN data used in this study were obtained from the field studies listed below. [...]. CCN concentrations in the studies were measured with a continuous-flow streamwise thermal gradient CCN counter (CCNC, models CCN-100 or CCN-200, DMT, Longmont, CO, USA). In the instrument, the supersaturation (S) was cycled through different S values, typically between 0.1 and 1.1 %, defined by controlled temperature gradients inside the CCNC column. Particles with a critical supersaturation $S_c > S$ in the column are activated and form water droplets. Droplets with diameters larger 1 μm are detected by an optical particle counter (OPC) at the exit of the column. Further information on the CCN measurements and instrumentation can be found elsewhere (refs). Note that the CCN measurements were conducted in size-resolved mode, which is one established approach to retrieve κ , as outlined in detail elsewhere (refs). This applies to all three main data sets used here for the κ_{org} and κ_{inorg} retrieval (i.e., ATTO, PRIDE-PRD-2006, and CARE-Beijing-2006, shown in Fig. 1B) as well as for most of data sets from previous field studies outlined below.”

[R1.10] Page 5, line 15 and following: It is argued why the method works even though dust and sea salt are not represented in the AMS data. This is likely a result of the upper size limit of the AMS, since

it has been shown that sea spray composition varies strongly with size. This size dependence is implied with the statement “accounts for most of the CCN”. Further, Table S2 explains the sizes that the data are specific to, which also limits the expected composition.

[A1.10] Thank you for pointing this out. We implemented this thought into the text on page 5 as follows:

“Note here that sea salt and mineral dust aerosol populations typically reach far into the supermicron size range, which implies that the AMS measurement with its upper cut-off size at about 600 nm (50 % transmission) excludes significant parts of both aerosol components.”

[R1.11] Page 6, lines 4-5 on statement “Changes in the aerosol hygroscopicity affect the growth of the particles in the subsaturated regime and therefore R_{Faci} as well as their activation to cloud droplets under supersaturated conditions and therefore R_{Faci} .”: Note that for organic compounds, it has frequently been documented that the hygroscopicity measured above water saturation – that is, deduced from observed CCN activity – is higher than that observed in the subsaturated regime, i.e., deduced from light scattering or HTDMA data. The authors should note this as a caveat to their methodology, in that the derived k_{org} is likely an upper estimate when used for direct forcing calculations. As seen in Table 1, the global mean value for direct effects is barely changed even with the higher k_{org} , but this may affect conclusions around local effects (Figure 2D). The authors could perhaps show, in the Supplement, the equivalents of Figure 2D for the second and third cases in Table 1 to address this possibility.

[A1.11] Important aspect – thanks for pointing this out. We clarified the text here by adding the following statement on page 8:

“Previous studies documented for organic compounds that the hygroscopicity measured above water saturation, as deduced from observed CCN activity, is higher than that observed in the subsaturated regime, as deduced from light scattering or hygroscopic tandem differential mobility analyser (HTDMA) data (Zheng et al., 2020; Mikhailov et al., 2021). This means that the κ_{org} derived here from CCN activity is likely an upper estimate when used for direct forcing calculations. Albeit, Table 1 suggests that the $\Delta\kappa$ between super- and subsaturated retrievals barely changes the global mean direct effects, it is still worth noting as it might affect conclusions on local effects as indicated by Fig. 2D.”
Zheng et al., Long-range transported North American wildfire aerosols observed in marine boundary layer of eastern North Atlantic, Environment International, 139, 2020.
Mikhailov et al., Water uptake of subpollen aerosol particles: hygroscopic growth, cloud condensation nuclei activation, and liquid–liquid phase separation, Atm. Chem. Phys, 21, 2021.

[R1.12] Page 8, lines 12-14 on statement “Second, the higher sensitivity of the model to changes in κ_{inorg} relative to changes κ_{org} stresses the importance of a correct representation of the abundance and hygroscopicity of the small number of relevant inorganic aerosol components (i.e., sulfates, nitrates, ammonium) in the models.”: the hygroscopicity of the inorganic components (when measured ions are interpreted as various salts, but excluding sodium compounds which are also excluded herein) does not vary that much, which is also stated on page 3, line 20 “These campaign-specific results show a larger range in κ_{org} ranging from 0.06 to 0.19 than for κ_{inorg} , ranging from 0.63 to 0.71 [...]”.

[A1.12] Referee 2 indicated in [R2.7] that not all numbers in Table 2 were updated appropriately, after more previous field campaigns were taken into account. We have corrected Table 2 – see [A2.7] – and the range of κ_{inorg} from previous campaigns is now 0.46 to 0.71, which is comparatively wide. Accordingly, the aforementioned statement seems still appropriate.

[R1.13] Just an observation: For a nonabsorbing aerosol, as assumed here, simulation of the size distribution can be an important factor, in order to select an appropriate mass scattering efficiency.

Composition also plays a significant role, of course. Interestingly, prior work showed that the ZSR mixing rule (the basis for the kappa mixing rule) also applies to scattering calculations and the efficiency curves could be weighted by the composition. Also interestingly, depending on how the calculation is set up, there is less dependence on the degree of neutralization of sulfate than might be expected. This is because the lower hygroscopicity of fully neutralized sulfate is balanced by the additional ammonium mass, while the lower ammonium mass per sulfate ion in bisulfate is compensated by the higher water content; variation in refractive index between those cases is a minor factor.

[A1.13] Thanks for this observation – interesting indeed and probably a starting point for a follow-up study.

[R1.14] If you would like to add a North American continental site, the following publication also used AMS and size-resolved CCN data in a two-component model to draw a similar conclusion ($k_{\text{org}} = 0.13$, assuming $k_{\text{inorg}} = 0.6$):

Levin, E. J. T., Prenni, A. J., Palm, B. B., Day, D. A., Campuzano-Jost, P., Winkler, P. M., Kreidenweis, S. M., DeMott, P. J., Jimenez, J. L., and Smith, J. N.: Size-resolved aerosol composition and its link to hygroscopicity at a forested site in Colorado, *Atmos. Chem. Phys.*, 14, 2657–2667, <https://doi.org/10.5194/acp-14-2657-2014>, 2014.

[A1.14] Thank you for this hint. We have included the study into the manuscript:

- The relevant references were included in several places in the manuscript:
 - o Levin et al.: An annual cycle of size-resolved aerosol hygroscopicity at a forested site in Colorado, *J. Geophys. Res.*, 117, D06201, 2012.
 - o Levin et al.: Size-resolved aerosol composition and its link to hygroscopicity at a forested site in Colorado, *Atm. Chem. Phys.*, 14, 2657-2667, 2014.
- The North American data set was added to Fig. 1C. The revised figure is shown below.

- A short description of the North American data set was added to the method section.
- Table S2 was updated in the supplementary information was updated.
- The new data set was also added to Fig. S5:

Responses to Referee #2

In this manuscript two globally representative average values of κ_{org} and κ_{inorg} are retrieved through a bivariate linear regression fit of extensive data sets of ϵ_{org} and κ from 15 field sites worldwide. The effective aerosol hygroscopicity (κ) is derived via global CCN measurement using a continuous-flow streamwise thermal gradient CCN counter, and the organic mass fraction (ϵ_{org}) is measured by aerosol mass spectrometers (AMS). The authors also investigate the sensitivity of radiative forcing to this pair values in the ECHAM-HAM aerosol-climate model, and conclude the sensitivity is small. Overall, this is a well written study, and the main results are clearly presented and discussed. I enjoyed reading this article and I think this manuscript will be appealing to modelers interested in aerosol-climate modelling. As such I believe this manuscript is well suited to Nature Communications, and I kindly ask the authors to carefully address (or consider) minor comments below before final publication.

Minor comments

[R2.1] One of the main contributions of this study is performing the retrieval of global representative κ_{org} and κ_{inorg} values. Can the authors further clarify the following questions about it? a) In Figure 1B, the linear regression is only based on data sets of three sites (i.e., Amazon, Beijing, and Guangzhou). Is it more suitable to do a retrieval by using more data from other sites that are included in Figure 1C? On page 12 lines 25-28, the authors list the sample size as one criterion for choosing the retrieval sample. We may lose representative data sets from other sites based on this criterion.

[A2.1] Many thanks for this comment. The criteria to include a data set into the retrieval are quite strict indeed. We decided to include only size-resolved CCN datasets with more than 100 data points, because smaller data sets might be strongly affected by outliers. The three data sets included are characterized by long-term measurements and a variability of the relevant parameters over a wide ϵ_{org} range, which are both important conditions to obtain statistically robust bivariate regression fits as outlined below. Furthermore, we also conducted the retrieval with all data sets and found that the results did not change significantly.

b) Further, the number of ATTO data points is much larger than those of the other two sites, does this mean the data represent more remote rain forest? This influence looks more obvious when we check details in Table S3. A merged κ_{org} is 0.12 ± 0.02 , which is close to $\kappa_{\text{org}} = 0.13 \pm 0.02$ at ATTO but almost twice as $\kappa_{\text{org}} = 0.06 \pm 0.03$ and 0.07 ± 0.03 at PRD and BEI, respectively.

The methods section on page 13 explains how the data was handled:

“For the three data sets with $n > 100$, the κ data was binned onto an ϵ_{org} grid, with ϵ_{org} increments of 0.01, and the data were averaged within each bin (see Fig. S4). The binned data from ATTO, PRD, and BEI were merged, which yielded our optimized retrieval (see Fig. S4 and Fig. S3). The binning ensures that the data across the entire ϵ_{org} range is weighted equally and that the regression is not biased by differences in the density of the original n data points.”

By binning the three data sets individually, the influence of different sample size is cancelled.

The fact that the ATTO data set has more influence on the κ_{org} than the PRD and BEI data, as the reviewer correctly pointed out, is not an effect of the different dataset size (as this is eliminated by the binning), but rather by the different ϵ_{org} being covered. Specifically, the ATTO aerosol has high organic fractions and accordingly most data points at high ϵ levels. In contrast, the BEI and PRD data has high inorganic fractions and therefore most data points at low ϵ levels. This means that κ_{org} can be retrieved from the ATTO data more reliable, compared to the BEI and PRD data, which require a rather ‘far’ extrapolation to $\epsilon_{\text{org}} = 1$ yielding κ_{org} .

c) The binned method successfully reduces the data density effect, but I am wondering if this will reduce the uncertainty of the linear regression correspondingly. Is that why the data points in Figure 1B have large uncertainties, but the uncertainty of the fit is so small? This discrepancy is also shown clearly in Figure S4. Is there a better way to present the uncertainty?

The uncertainty of the fit is not directly related to the binning method. The error can also be bigger after binning, which is not the case in this study because the linear relationship describes the data very well after binning and this is the reason for the low error in the fit. The result of the fit is different with and without binning; depending on the scientific question the result with or without binning is better.

For further clarification, we have added the following paragraph to the section on the retrieval to clarify those points:

“Note that a binning and fitting of all data sets, compared to an averaging and fitting of the binned three largest data sets only (see above), did not yield a significantly different κ_{org} and κ_{inorg} retrieval within the range of the uncertainty. This shows that the optimized retrieval is statistically robust, independent of whether the smaller data sets are included or not. Binning all data sets with comparable counting error and averaging the binned results afterwards, however, seems to be the statistical most valid approach at this point.”

[R2.2] Page 9 line 16. The section of “Field studies and data sets” is a bit long. Adding a large table similar to Table S2 may help us to quickly grab campaign information (e.g., date, geographical location, data sources and special conditions) and make comparisons readily. For different stations, a summary of particle size distribution, chemical composition and potential particle sources may help to show how representative the dataset is. I feel the current section generally provides fragmented information though it lists all sites and related information. Also, readers may want to see how the campaign data sets are effectively integrated and to judge the validity of results without having to consult another paper. I would suggest the authors add some critical information (e.g., the sample size of different stations in Fig. 1C, more detailed data filtering method, and visualized CCN and AMS data from important stations) in SI, and this will help to fill the gap between campaign results and the paper conclusion.

[A2.2] Thanks you for the suggestion. We have added a table as suggested by the referee with the most essential information, but also kept the text section in its current form for more detailed information.

[R2.3] On page 12 line 28-30, the authors mention “the AMS and CCN counter measurements are most comparable for the largest particle diameters measured at the lowest supersaturation (i.e., $S \sim 0.1\%$), which corresponds to accumulation mode particles.” Will this method of data selection bring some bias? Additionally, the data sets for the retrieval were obtained in 2006 and in 2014-2015, and we know the AMS was updated frequently during these years, so I am just curious if the update will affect the data analysis process.

[A2.3] To clarify this aspect, we introduced the following statement into page 15:

“Note here that κ is typically size-dependent and, therefore, focussing on one particular size range can introduce some bias. At the same time, κ values are typically rather constant across a given size mode (Rose et al., 2010; Pöhlker et al., 2016). Accordingly, the approach chosen here can be regarded as representative for accumulation mode particles as the prime source of CCN.”

We do not have any indications that the AMS data sets from 2006 and 2014/15 might be not comparable due to instrumental updates.

[R2.4] Page 5 lines 29-30 the authors write, “the assumption of an internally mixed aerosol as an average of the global composition is justified (31).” In contrast, recent studies showed that neither external nor internal mixtures fully represent the mixing state of atmospheric aerosols. It would be appropriate for the authors to provide a bit more discussion as to the validity of their assumption or at least list some potential caveats. Similarly, the organic coating and its effect on surface tension may also raise questions while assuming the surface tension of pure water on page 2 line 28.

[A2.4] This is a valid objective. We modified and extended the original statement:

“In principle, the retrieval might differ for freshly emitted, externally mixed aerosol particles that have distinct compositions. Since aerosol populations tend to become internally mixed within tens of kilometers or few hours away from emission sources, however, the assumption of an internally mixed aerosol as an average of the global composition is justified (ref).”

as follows

“This mixing state of the aerosol components affects κ_{org} and κ_{inorg} and its representation in models is associated with large uncertainties (Zhuo et al., 2018; Zheng et al., 2021; Kodros et al., 2018). Generally, an aerosol population tends to become more and more internally mixed with atmospheric residence time, even within tens of kilometers or after few hours of transport (Ervens et al., 2010). At the same time, studies suggest that the mixing state of atmospheric aerosols is neither fully represented by the assumption or parameterization of an external mixture nor internal mixture (Sharma et al., 2018; Body et al., 2018). Our parameterization of κ_{org} and κ_{inorg} is based on a wide variety of atmospheric conditions at different locations and, therefore, represents an atmospherically relevant distribution of prevalent aerosol mixing states. At the same time, special aerosol populations, such as freshly emitted and externally mixed particles, might deviate from the average retrieval.”

[R2.5] One implication may underestimate the significance of some studies on aerosol hygroscopicity. On page 8 lines 14-16 the authors write, “In turn, this also implies that an experimental determination of the κ of individual compounds within the thousands of different organics – with many of them being unidentified – is not essential to reduce uncertainties in the effective radiative forcing of aerosol particles.” I agree that it is not essential to further examine thousands of different organics. However, the variation of the κ_{org} has been represented using the atomic O:C ratio that reflects the oxidation level of the organics.^{6, 7} Maybe the authors could mention this research direction for future study since this could probably fill the gap between data of Vogel et al. (and maybe Cerully et al.) and your fitted line in Fig. S6 in the future?

[A2.5] We added the following statement on page 9, following the referee’s suggestion:

“The variability of κ within the organic aerosol fraction might be efficiently represented by average chemical properties such as the oxygen-to-carbon ration (Jimenez et al., 2009; Cerulli et al., 2015).”

[R2.6] On page 6 line 3: In the section of “Climate model sensitivity to changes in κ_{org} and κ_{inorg} ”, the authors describe the great details of absolute and relative changes in radiative forcing, and the sensitivity of radiative forcing to κ_{org} and κ_{inorg} . Readers may want to see more in-depth discussion and implications. For example, on page 6 line 15: “Further note that the model yields an RF_{ari} of -0.11Wm^{-2} (Table 1), which is about half of the value of -0.22W m^{-2} given in the recent report of the Intergovernmental Panel on Climate Change (IPCC) (2).” It is a good comparison here but what does this imply and is it a big difference? Another example on page 8 line 5: “Note here the much larger uncertainty in ER_{Faer} ”. Where does the large uncertainty come from, and will it influence the conclusion? Lastly, I am also wondering if there are some available sensitivity studies by applying some different models, and if these results were comparable to results here?

[A2.6] We modified the original statement:

“Further note that the model yields an RF_{ari} of -0.11Wm^{-2} (Table 1), which is about half of the value of -0.22W m^{-2} given in the recent report of the Intergovernmental Panel on Climate Change (IPCC) (2).”

as follows and also added some more explanation and context. Note also that we changed the RF_{ari} value according to the latest IPCC report.

“Further note the ΔRF_{ari} and ΔER_{Faer} from ECHAM-HAM can be considered rather as lower limit values and corresponding results from other climate models might be higher. Myhre et al. (45) showed in a model comparison study that ECHAM-HAM yields a global mean anthropogenic RF (all-sky) of $-0.15 W m^{-2}$, in comparison to the 16-model mean of $-0.27 \pm 0.15 W m^{-2}$ (\pm one standard deviation). Along these lines, the RF_{ari} of $\sim -0.11 W m^{-2}$ derived here from ECHAM-HAM (Table 1) is much less than the value of $-0.3 W m^{-2}$ given in the recent report of the Intergovernmental Panel on Climate Change (IPCC) (2).”

Typos and Formats

[R2.7] Page 3 lines 19-20: “These campaign-specific results show a larger range in κ_{org} ranging from 0.06 to 0.19 than for κ_{inorg} , ranging from 0.63 to 0.71...” The ranges of κ_{org} and κ_{inorg} summarized here are not consistent with the original data in Table S2. Please correct or clarify it correspondingly.

[A2.7] True. Obviously, the numbers in the text were not finally updated in the course of several revisions of Table S2. We have corrected the statement as follows:

“These campaign-specific results show large ranges in κ_{org} ranging from 0.04 to 0.25 than for κ_{inorg} , ranging from 0.46 to 0.71...”

[R2.8] Page 11 lines 15-17: The work by Cerully et al. (55) is referred to SI Fig.S6, “All campaign data sets from Fig. 1C shown here in gray, in combination with data from Vogel et al. (32) in blue and Cerully et al. (55) in red.” The red markers are missing in this figure.

[A2.8] Thank you – by mistake indeed an older version of the figure was included. We replaced it by the latest version of the figure shown below:

[R2.9] For the Figure 1C on page 4, can the authors at least refer these markers to Fig.S5 so that we know which site each of them represents? On page 5 lines 8-17, we may also need to refer the discussions to Fig. S5.

[A2.9] Colors and shapes of the markers are identical in Fig. 1A, 1C and S5. To underline this, we added the following statement to the caption of Fig. 1: “The colors and shapes of the markers in panels (A) and (C) as well as in Fig. S5 are identical to relate the data sets to the corresponding sites.”

Moreover, we added the following sentence to page line 8ff. as suggested by the referee:

“The individual data sets are shown in Fig. S5.”

[R2.10] In SI Table S2, the retrieved κ_{org} is 0.19 at PRD station in Rose et al. But for the same PRD station, the retrieved κ_{org} is 0.06 in this study in Table S3. Can the authors explain the reason why they are quite different since some readers may be interested in this?

[A2.10] Correct, the original retrieval from Rose et al., 2011 in Table S2 and the retrieval with the method of this study in Table S3 are not identical. This can be explained by the different methods used for the retrieval: The most important differences are that (i) the data sets were binned on the ε_{org} axis in this study and (ii) that a bivariate linear fit was used here (see experimental section). To clarify this aspect, the following statement was added to the caption of Table S3:

“Note that the retrieved κ_{org} and κ_{inorg} in this table were obtained with the approach as outlined in the experimental section and deviate from the retrievals in the original studies as summarized Table S2. The main differences are the binning on the ε_{org} axis and the application of the linear bivariate fit applied here.”

[R2.11] Page 5 lines 8: The description “robustly and broadly” is a qualitative statement and readers may wonder how statistically representative the relationship is. Therefore, is it reasonable here to do some significance tests and see if it’s statistically significant?

[A2.11] As outlined in the method section, various variants of the analysis and retrieval gave pretty much the same results, which was a strong indication for us that the overall approach is robust indeed.

[R2.12] Page 6 line 9: “The model parameterizes the overall κ through Eq. 3 based on compound-specific κ values for sea salt, black carbon, mineral dust, organic carbon, and sulfate aerosols (details in supplementary materials).” No data information is available in SI. What are the “details” mean here?

[A2.12] In the course of the submission, the experimental section was moved back and forth between the SI and the appendix of the manuscript. The “details” including references for further reading are in fact now in the experimental sections and not in the SI, as the referee pointed out correctly here. Thanks for providing this hint. We modified the corresponding statement as follows: *“The model parameterizes the overall κ through Eq. 3 based on compound-specific κ values for sea salt, black carbon, mineral dust, organic carbon, and sulfate aerosols (details in experimental section).”*

References

1. Kodros, J. K.; Hanna, S. J.; Bertram, A. K.; Leaitch, W. R.; Schulz, H.; Herber, A. B.; Zanutta, M.; Burkart, J.; Willis, M. D.; Abbatt, J. P. D.; Pierce, J. R., Size-resolved mixing state of black carbon in the Canadian high Arctic and implications for simulated direct radiative effect. *Atmos. Chem. Phys.* 2018, 18 (15), 11345-11361.
2. Sharma, N.; China, S.; Bhandari, J.; Gorkowski, K.; Dubey, M.; Zaveri, R. A.; Mazzoleni, C., Physical Properties of Aerosol Internally Mixed With Soot Particles in a Biogenically Dominated Environment in California. *Geophys. Res. Lett.* 2018, 45 (20), 11473-11482.
3. Zheng, Z. H.; West, M.; Zhao, L.; Ma, P. L.; Liu, X. H.; Riemer, N., Quantifying the structural uncertainty of the aerosol mixing state representation in a modal model. *Atmos. Chem. Phys.* 2021, 21 (23), 17727-17741.
4. Zhou, C.; Zhang, H.; Zhao, S. Y.; Li, J. N., On Effective Radiative Forcing of Partial Internally and Externally Mixed Aerosols and Their Effects on Global Climate. *J. Geophys. Res.-Atmos.* 2018, 123 (1), 401-423.
5. Bondy, A. L.; Bonanno, D.; Moffet, R. C.; Wang, B. B.; Laskin, A.; Ault, A. P., The diverse chemical mixing state of aerosol particles in the southeastern United States. *Atmos. Chem. Phys.* 2018, 18 (16), 12595-12612.
6. Jimenez, J. L.; Canagaratna, M. R.; Donahue, N. M.; Prevot, A. S.; Zhang, Q.; Kroll, J. H.; DeCarlo, P. F.; Allan, J. D.; Coe, H.; Ng, N. L.; Aiken, A. C.; Docherty, K. S.; Ulbrich, I. M.; Grieshop, A. P.; Robinson, A. L.; Duplissy, J.; Smith, J. D.; Wilson, K. R.; Lanz, V. A.; Hueglin, C.; Sun, Y. L.; Tian, J.; Laaksonen, A.; Raatikainen, T.; Rautiainen, J.; Vaattovaara, P.; Ehn, M.; Kulmala, M.; Tomlinson, J. M.; Collins, D. R.; Cubison, M. J.; Dunlea, E. J.; Huffman, J. A.; Onasch, T. B.; Alfarra, M. R.; Williams, P. I.; Bower, K.; Kondo, Y.; Schneider, J.; Drewnick, F.; Borrmann, S.; Weimer, S.; Demerjian, K.; Salcedo, D.; Cottrell, L.; Griffin, R.; Takami, A.; Miyoshi, T.; Hatakeyama, S.; Shimono, A.; Sun, J. Y.; Zhang, Y. M.; Dzepina, K.; Kimmel, J. R.; Sueper, D.; Jayne, J. T.; Herndon, S. C.; Trimborn, A. M.; Williams, L. R.; Wood, E. C.; Middlebrook, A. M.; Kolb, C. E.; Baltensperger, U.; Worsnop, D. R., Evolution of organic aerosols in the atmosphere. *Science* 2009, 326 (5959), 1525-9.
7. Cerully, K. M.; Bougiatioti, A.; Hite, J. R.; Guo, H.; Xu, L.; Ng, N. L.; Weber, R.; Nenes, A., On the link between hygroscopicity, volatility, and oxidation state of ambient and water-soluble aerosols in the southeastern United States. *Atmos. Chem. Phys.* 2015, 15 (15), 8679-8694.

REVIEWERS' COMMENTS

Reviewer #1 (Remarks to the Author):

The authors have done a thorough job responding to the reviews; thank you for considering them carefully. I recommend publication. The only further suggestion I would make is as follows.

“Previous studies documented for organic compounds that the hygroscopicity measured above water saturation, as deduced from observed CCN activity, is higher than that observed in the subsaturated regime, as deduced from light scattering or hygroscopic tandem differential mobility analyser (HTDMA) data (Zheng et al., 2020; Mikhailov et al., 2021). This means that the k_{org} derived here from CCN activity is likely an upper estimate when used for direct forcing calculations. Albeit, Table 1 suggests that the Δk between super- and subsaturated retrievals barely changes the global mean direct effects, it is still worth noting as it might affect conclusions on local effects as indicated by Fig. 2D.”

Zheng et al., Long-range transported North American wildfire aerosols observed in marine boundary layer of eastern North Atlantic, *Environment International*, 139, 2020.

Mikhailov et al., Water uptake of subpollen aerosol particles: hygroscopic growth, cloud condensation nuclei activation, and liquid–liquid phase separation, *Atm. Chem. Phys*, 21, 2021.

These are interesting and appropriate papers to make this point. I had in mind earlier lab studies that explored nonidealities of secondary organic aerosol components which was observed early on in the study of SOA hygroscopicity. There are a number of such studies that can be added; here is one suggestion for a paper studying this issue that specifically mentions the “gap” in the article title:

M.D. Petters, H. Wex, C.M. Carrico, E. Hallbauer, A. Massling, G.R. McMeeking, L. Poulain, Z. Wu, S.M. Kreidenweis, F. Stratmann: Towards closing the gap between hygroscopic growth and activation for secondary organic aerosol – Part 2: Theoretical approaches. *Atmos. Chem. Phys.*, 9 (2009), pp. 3999-4009, 10.5194/acp-9-3999-2009

One additional note: Petters and Kreidenweis (2007) is listed as both reference 7 and 78.

Kind regards, Sonia Kreidenweis

Responses to the second round of review of the manuscript “Global organic and inorganic aerosol hygroscopicity and its effect on radiative forcing”, by Pöhlker et al., submitted for publication in Nature Communications

Dear editor, we would like to thank you and referee 1 again for the valuable comments and useful suggestions to improve our manuscript. Below, you will find our responses to the individual comments of the referees along with a description of changes that we did in the manuscript. We have used the following color code:

- **Black: the referee’s comments.**
- **Blue: the authors’ responses.**
- **Red: Quotes from the original manuscript.**
- **Red & italic: Text modifications in the manuscript related to the referee comments.**

Responses to Referee #1 (Sonia Kreidenweis)

The authors have done a thorough job responding to the reviews; thank you for considering them carefully. I recommend publication.

We thank Sonia Kreidenweis for the thorough second round of reviewing our manuscript and the positive evaluation.

[R1.1] The only further suggestion I would make is as follows.

“Previous studies documented for organic compounds that the hygroscopicity measured above water saturation, as deduced from observed CCN activity, is higher than that observed in the subsaturated regime, as deduced from light scattering or hygroscopic tandem differential mobility analyser (HTDMA) data (Zheng et al., 2020; Mikhailov et al., 2021). This means that the κ_{org} derived here from CCN activity is likely an upper estimate when used for direct forcing calculations. Albeit, Table 1 suggests that the $\Delta\kappa$ between super- and subsaturated retrievals barely changes the global mean direct effects, it is still worth noting as it might affect conclusions on local effects as indicated by Fig. 2D.”

Zheng et al., Long-range transported North American wildfire aerosols observed in marine boundary layer of eastern North Atlantic, *Environment International*, 139, 2020.

Mikhailov et al., Water uptake of subpollen aerosol particles: hygroscopic growth, cloud condensation nuclei activation, and liquid–liquid phase separation, *Atm. Chem. Phys.*, 21, 2021.

These are interesting and appropriate papers to make this point. I had in mind earlier lab studies that explored nonidealities of secondary organic aerosol components which was observed early on in the study of SOA hygroscopicity. There are a number of such studies that can be added; here is one suggestion for a paper studying this issue that specifically mentions the “gap” in the article title:

M.D. Petters, H. Wex, C.M. Carrico, E. Hallbauer, A. Massling, G.R. McMeeking, L. Poulain, Z. Wu, S.M. Kreidenweis, F. Stratmann: Towards closing the gap between hygroscopic growth and activation for secondary organic aerosol – Part 2: Theoretical approaches. *Atmos. Chem. Phys.*, 9 (2009), pp. 3999-4009, 10.5194/acp-9-3999-2009

[A1.1] Thank you for your thoughts here. We have added the reference you suggested (Petters et al., 2009) and kept the two references that we selected already in the course of the first revision (Zheng et al., 2020; Mikhailov et al., 2021).

[R1.2] One additional note: Petters and Kreidenweis (2007) is listed as both reference 7 and 78.

[A1.2] Thank you for pointing this out. It has been corrected in the revised manuscript file.